# A Regression Approach to Learning-Augmented Online Algorithms

**Keerti Anand**
Department of Computer Science
Duke University
Durham, NC 27705
kanand@cs.duke.edu

**Rong Ge**
Department of Computer Science
Duke University
Durham, NC 27705
rongge@cs.duke.edu

**Amit Kumar**
Department of Computer Science
Indian Institute of Technology
Hauz Khas, New Delhi - 110016
amitk@cse.iitd.ernet.in

**Debmalya Panigrahi**
Department of Computer Science
Duke University
Durham, NC 27705
debmalya@cs.duke.edu

## Abstract

The emerging field of learning-augmented online algorithms uses ML techniques to predict future input parameters and thereby improve the performance of online algorithms. Since these parameters are, in general, real-valued functions, a natural approach is to use regression techniques to make these predictions. We introduce this approach in this paper, and explore it in the context of a general online search framework that captures classic problems like (generalized) ski rental, bin packing, minimum makespan scheduling, etc. We show nearly tight bounds on the sample complexity of this regression problem, and extend our results to the agnostic setting. From a technical standpoint, we show that the key is to incorporate online optimization benchmarks in the design of the loss function for the regression problem, thereby diverging from the use of off-the-shelf regression tools with standard bounds on statistical error.

## 1 Introduction

A recent trend in *online algorithms* has seen the use of *future predictions* generated by ML techniques to bypass pessimistic worst-case lower bounds. A growing body of work has started to emerge in this area in the last few years addressing a broad variety of problems in online algorithms such as rent or buy, caching, metrical task systems, matching, scheduling, experts learning, stopping problems, and others (see related work for references). The vast majority of this literature is focused on *using* ML predictions in online algorithms, but does not address the question of how these predictions are generated. This raises the question: *what can we learn from data that will improve the performance of online algorithms?* Abstractly, this question comes in two inter-dependent parts: the first part is a *learning* problem where we seek to learn a function that maps the feature domain to predicted parameters, and the second part is to re-design the online algorithm to use these predictions. In this paper, we focus on the first part of this design pipeline, namely *we develop a regression approach to generating ML predictions for online algorithms*.

35th Conference on Neural Information Processing Systems (NeurIPS 2021).

Before delving into this question further, we note that there has been some recent research that focuses on the learnability of predicted parameters in online algorithms. Recently, Lavastida *et al.* [33], building on the work of Lattanzi *et al.* [32], took a data-driven algorithms approach to design online algorithms for scheduling and matching problems via *learned weights*. In this line of work, the goal is to observe sample inputs in order to learn a set of weights that facilitate better algorithms for instances from a fixed distribution. In contrast, Anand *et al.* [4] relied on a classification learning approach for the Ski Rental problem, where they aimed to learn a function that maps the feature set to a binary label characterizing the optimal solution. But, in general, the value of the optimal solution is a real-valued function, which motivates a regression approach to learning-augmented online algorithms that we develop in this paper.

To formalize the notion of an unknown optimal solution that we seek to learn via regression, we use the *online search* (ONLINESEARCH) framework. In this framework, there is as an input sequence $\Sigma = \sigma_1, \sigma_2, \ldots$ available offline, and the actual online input is a prefix of this sequence $\Sigma_T = \sigma_1, \sigma_2, \ldots, \sigma_T$, where the length of the prefix $T$ is revealed online. Namely, in each online step $t > 0$, there are two possibilities: either the sequence *ends*, i.e., $T = t$, or the sequence *continues*, i.e., $T > t$. The algorithm must maintain, at all times $t$, a solution that is feasible for the current sequence, i.e., for the prefix $\Sigma_t = \sigma_1, \ldots, \sigma_t$. The goal is to obtain a solution that is of minimum cost among all the feasible solutions for the actual input sequence $\Sigma_T$.

We will discuss applicability of the ONLINESEARCH framework in more detail in Section 1.2, but for a quick illustration now, consider the ski rental problem in this framework. In this problem, if the sequence continues on day $t$, then the algorithm must rent skis if it has not already bought them. In generalizations of the ski rental problem to multiple rental options, the requirement is that one of the rental options availed by the algorithm must cover day $t$. We will show in Section 1.2 that we can similarly model several other classic online problems in the ONLINESEARCH framework.

We use the standard notion of *competitive ratio*, defined as the worst case ratio between the algorithm's cost and the optimal cost, to quantify the performance of an online algorithm. For online algorithms with predictions, we follow the terminology in [41] that is now standard: we say that the *consistency* and *robustness* of an algorithm are its competitive ratios for correct predictions and for arbitrarily incorrect predictions respectively. Typically, we fix consistency at $1 + \epsilon$ for a hyper-parameter $\epsilon$ and aim to minimize robustness as a function of $\epsilon$.

We make some mild assumptions on the problem. First, we assume that solutions are *composable*, i.e., that adding feasible solutions for subsequences ensures feasibility over the entire sequence; second, that cost is *monotone*, i.e., the optimal cost for a subsequence is at most that for the entire sequence; and third, that the offline problem is (approximately or exactly) *solvable*. These assumptions hold for essentially all online problems we care for.

## 1.1  Our Contributions

As a warm up, we first give an algorithm called DOUBLE for the ONLINESEARCH problem *without predictions* in Section 2. The DOUBLE algorithm has a competitive ratio of 4. We build on the DOUBLE algorithm in Section 3, where we give an algorithm called PREDICT-AND-DOUBLE for the ONLINESEARCH problem *with predictions*. We show that the PREDICT-AND-DOUBLE algorithm has a consistency of $1 + \epsilon$ and robustness of $O(1/\epsilon)$, for any hyper-parameter $\epsilon > 0$. We also show that this tradeoff between consistency and robustness is asymptotically tight.

Our main contributions are in Section 4. In this section, we model the question of obtaining a learning-augmented algorithm for the ONLINESEARCH problem in a regression framework. Specifically, we assume that the input comprises a feature vector $x$ that is mapped by an unknown real-valued function $f$ to an input for the ONLINESEARCH problem $z$. In the training phase, we are given a set of labeled samples of the form $(x, z)$ from some (unknown to the algorithm) data distribution $\mathbb{D}$. The goal of the learning algorithm is to produce a mapping from the feature space to algorithmic strategies for the ONLINESEARCH problem, such that when it gets an unlabeled (test) sample $x$ from the same distribution $\mathbb{D}$, the algorithmic strategy corresponding to $x$ obtains a competitive solution for the actual input $z$ in the test sample (that is unknown to the algorithm).

The learning algorithm employs a regression approach in the following manner. It assumes that the function $f$ is from a hypothesis class $\mathcal{F}$, and obtains an empirical minimizer in $\mathcal{F}$ for a carefully crafted loss function on the training samples. The design of this loss function is crucial since a bound

on this loss function is then shown to translate to a bound on the competitive ratio of the algorithmic strategy. (Indeed, we will show later that because of this reason, standard loss functions used in regression are inadequate for our purpose.) Finally, we use statistical learning theory for real-valued functions to bound the sample complexity of the learner that we designed.

Using the above framework, we show a sample complexity bound of $O\left(\frac{H \cdot d}{\epsilon}\right)$ for obtaining a competitive ratio of $1 + \epsilon$, where $H$ and $d$ respectively represent the log-range of the optimal cost and a measure of the expressiveness of the function class $\mathcal{F}$ called its pseudo-dimension.[1] We also extend this result to the so-called agnostic setting, where the function class $\mathcal{F}$ is no longer guaranteed to contain an exact function $f$ that maps $x$ to $z$, rather the competitive ratio is now in terms of the *best* function in this class that approximates $f$. We also prove nearly matching lower bounds for our sample complexity bounds in the two models.

Our framework can also be extended to the setting where the offline optimal solution is hard to compute, but there exists an algorithm with competitive ratio $c$ given the *cost* of optimal solution. In that case our algorithms gives a competitive ratio $c(1 + \epsilon)$, which can still be better than the competitive ratio without predictions (see examples in next subsection).

## 1.2 Applicability of the ONLINESEARCH framework

The ONLINESEARCH framework is applicable *whenever an online algorithm benefits from knowing the optimal value of the solution*. Many online problems benefit from this knowledge, which is sometimes called *advice* in the online algorithms literature. For concreteness, we give three examples of classic problems – *ski rental with multiple options*, *online scheduling*, and *online bin packing* – to illustrate the applicability of our framework. Our algorithm PREDICT-AND-DOUBLE (explained in more detail in section 3) successively predicts the optimal value of the solution and appends the corresponding solution to its output.

**Ski Rental with Multiple Options.** Generalizations of the ski rental problem with multiple options have been widely studied (e.g., [1, 34, 37, 19]), recently with ML predictions [44]. Suppose there are $V$ options (say coupons) at our disposal, where coupon $i$ costs us $C_i$ and is valid for $d_i$ number of days. Given such a setup, we need to come up with a schedule: $\{(t_k, i_k), k = 1, 2 \ldots\}$ that instructs us to buy coupon $i_k$ at time $t_k$. (The classic ski rental problem corresponds to having only two coupons $C_1 = 1, d_1 = 1$ and $C_2 = B, d_2 \to \infty$.) Our ONLINESEARCH framework is applicable here: a solution that allows us to buy coupons valid time $t$ is also a valid solution for all times $s \leq t$. Further, PREDICT-AND-DOUBLE can be implemented efficiently as we can compute OPT$(t)$, for any time $t$ using a dynamic program.

**Online Scheduling.** Next, we consider the classic online scheduling problem where the goal is to assign jobs arriving online to a set of identical machines so as to minimize the maximum load on any machine (called the *makespan*). For this algorithm, the classic list scheduling algorithm [26] has a competitive ratio of 2. A series of works [23, 14, 29, 2] improved the competitive ratio to 1.924, and currently the best known result has competitive ratio of (approx) 1.92 [20]; in fact, there are nearly matching lower bounds [25]. However, if the optimal make-span (OPT) is *known*, then these lower bounds can be overcome, and a significantly better competitive ratio of 1.5 can be obtained in this setting [17] (see also [10, 31, 21, 22]). The ONLINESEARCH framework is applicable here with a slight modification: whenever PREDICT-AND-DOUBLE tries to buy a solution corresponding to a predicted value of OPT, we execute the 1.5-approximation algorithm based on this value. The problem still satisfies the property that a solution for $t$ jobs is valid for any prefix. We get a competitive ratio of $1.5 + O(\epsilon)$ that significantly outperforms the competitive ratio of 1.92 without predictions.

**Online Bin Packing.** As a third example, we consider the online bin packing problem. In this problem, items arrive online and must be packed into fixed-sized bins, the goal being to minimize the number of bins. (We can assume w.l.o.g., by scaling, that the bins are of unit size.) Here, it is known that the critical parameter that one needs to know/predict is not OPT but the number of items of *moderate* size, namely those sized between $1/2$ and $2/3$. If this is known, then there is a simple 1.5-competitive algorithm [5], which is not achievable without this additional knowledge. Again, our ONLINESEARCH framework can be used to take advantage of this result. In this case, the application is not as direct, because predicting OPT does not yield the better algorithm. Nevertheless,

---

[1]Intuitively, the notion of pseudo-dimension extends that of the well-known VC dimension from binary to real-valued functions.

an inspection of the algorithm in [5] reveals the following strategy: The items are partitioned into three groups. The items of size $\geq 2/3$ are assigned individual bins, items of size between $1/3$ and $1/2$ are assigned separate bins where at least two of them are assigned to each bin, and the remaining items are assigned a set of common bins. Clearly, the first two categories can be handled online without any additional information; this means that we can define a surrogate OPT (call it OPT$'$) that only captures the optimal number of bins for the common category. Note that the of prediction of OPT' serves as a substitute for knowing the numbers of items of moderate size. Now, if OPT$'$ is known, then we can recover the competitive ratio of $3/2$ by using a simple greedy strategy. This now allows us to use the ONLINESEARCH framework where we predict OPT$'$. As earlier, the ONLINESEARCH framework can be applied with slight modification: whenever PREDICT-AND-DOUBLE tries to buy a solution corresponding to a predicted value of OPT$'$, we execute the 1.5-competitive algorithm based on this value. The problem still satisfies the property that a solution for $t$ items is valid for any prefix.

## 1.3 Motivation for a cognizant loss function

In this work, we explore the idea of a carefully crafted loss function that can help in making better predictions for the online decision task. To illustrate this, consider the problem of balancing the load between machines/clusters in a data center where remote users are submitting jobs. The goal is to minimize the maximum load on any machine, also called the makespan of the assignment. The optimal makespan, which we would like to predict, depends on the workload submitted by individual users who are currently active in the system. Therefore, we would like to use the user features to predict their behavior in terms of the workload submitted to the server. A typical feature vector would then be a binary vector encoding of the set of users currently active in the system, and based on this information, a learning model trained on historical behavior of the users can predict (say) a histogram of loads that these users are expected to submit, and therefore, the value of the optimal makespan. The feature space can be richer, e.g., including contextual information like the time of the day, day of the week, etc. that are useful to more accurately predict user behavior. Irrespective of the precise learning model, the main idea in this paper is that the learner should try to optimize for competitive loss instead of standard loss functions. This is because the goal of the learner is not to accurately predict the workload of each user, but to eventually obtain the best possible makespan. For instance, a user who submits few jobs that are inconsequential to the eventual makespan need not be accurately predicted. Our technique automatically makes this adjustment in the loss function, thereby obtaining better performance on the competitive ratio.

## 1.4 Related Work

There has been considerable recent work in incorporating ML predictions in online algorithms. Some of the problems include: auction pricing [36], ski rental [41, 24, 4, 13, 44, 6], caching [35, 42, 28, 45], scheduling [41, 32, 39], frequency estimation [27], Bloom filters [38], online linear optimization [16], speed scaling [11], set cover [12], bipartite and secretary problems [9], etc. While most of these papers focus on designing online algorithms for ML predictions but not on the generation of these predictions, there has also been some work on the design of predictors using a binary classification approach [4]), and on the formal learnability of the predicted parameters [32, 33]. In contrast, we use a regression approach to the problem in this paper.

The PAC learning framework was first introduced by [43] in the context of learning binary classification functions, and related the sample complexity to the VC dimension of the hypothesis class. This was later extended to real-valued functions by [40], who introduced the concept of pseudo-dimension, and [30] (see also [15]), who introduced the fat shattering dimension, as generalizations of VC dimension to real-valued functions. For a comprehensive discussion of the extension of VC theory to learning real-valued functions, the reader is referred to the excellent text by [8]. A different approach was proposed by [3] (see also [7]) who analysed a model of learning in which the error of a hypothesis is taken to be the expected squared loss, and gave uniform convergence results for this setting. In this paper, we use pseudo-dimension and corresponding sampling complexity bounds in quantifying the complexity of the regression learning problem of predicting input length.

## 2 ONLINESEARCH **without Predictions**

As a warm up, we first describe a simple algorithm called DOUBLE (Algorithm 1) for the ONLI-NESEARCH problem *without predictions*. This algorithm places milestones on the input sequence corresponding to inputs at which the cost of the optimal solution doubles. When the input sequence crosses such a milestone, the algorithm buys the corresponding optimal solution and adds it to the existing online solution. This simple algorithm will form a building block for the algorithms that we will develop later in the paper; hence, we describe it and prove its properties below.

First, we introduce some notation.

**Definition 1.** *We use* OPT$(t)$ *to denote an optimal (offline) solution for the input prefix of length* $t$*; we overload notation to denote the cost of this solution by* OPT$(t)$ *as well.*

**Definition 2.** *Given an input length* $\tau$ *and any* $\alpha > 0$*, we use* MIN-LENGTH$(\alpha, \tau)$ *to denote the smallest length* $t$ *such that* OPT$(t) \geq \alpha \cdot$ OPT$(\tau)$*. The monotonicity property of* OPT *ensures that* MIN-LENGTH$(\alpha, \tau) > \tau$ *if* $\alpha > 1$*, and* MIN-LENGTH$(\alpha, \tau) \leq \tau$ *otherwise.*

---

**Algorithm 1** DOUBLE

**Input:** The input sequence $\mathcal{I}$.
**Output:** The online solution SOL.

Set $i := 0$, $\tau_0 := 1$, SOL $:= \emptyset$.
**for** $t = 1, 2, \ldots, T$
    **if** $t = \tau_i$
        Set $\tau_{i+1} =$ MIN-LENGTH$(2, \tau_i)$.
        Add OPT$(\tau_{i+1} - 1)$ to SOL.
        Increment $i$.

---

**Theorem 1.** *The* DOUBLE *algorithm is* 4*-competitive for the* ONLINESEARCH *problem.*

## 3 ONLINESEARCH **with Predictions**

In the previous section, we described a simple online algorithm for the ONLINESEARCH problem. Now, we build on this algorithm to take advantage of ML predictions. For now, we do not concern ourselves with how these predictions are generated; we will address this question in the next section.

Suppose we have a prediction $\hat{T}$ for the input length $T$ of an ONLINESEARCH problem instance. Naïvely, we might trust this prediction completely and buy the solution OPT$(\hat{T})$. While this algorithm is perfect if the prediction is accurate, it can fail in two ways if the prediction is inaccurate: (a) if $T \ll \hat{T}$ and therefore OPT$(T) \ll$ OPT$(\hat{T})$, then the algorithm has a large competitive ratio, and (b) if $T > \hat{T}$, then OPT$(\hat{T})$ may not even be feasible for $T$. A natural idea is to then progressively add OPT$(t)$ solutions for small values of $t$ (similar to DOUBLE) until a certain threshold is reached, before buying the predicted optimal solution OPT$(\hat{T})$. Next, if $T > \hat{T}$, the algorithm can resume buying solution OPT$(t)$ for $t > \hat{T}$, again using DOUBLE, until the actual input $T$ is reached.

One problem with this strategy, however, is that the algorithm does not degrade gracefully around the prediction, a property that we will need later in the paper. In particular, if $T$ is only slighter larger than $\hat{T}$, then the algorithm adds a solution that has cost $2 \cdot$ OPT$(\hat{T})$, thereby realizing the worst case scenario in Theorem 1 that was achieved without any prediction. Our work-around for this issue is to buy OPT$(t)$ for a $t$ slightly larger than $\hat{T}$, instead of OPT$(\hat{T})$ itself, which secures us against the possibility of the actual input being slightly longer than the prediction. We call this algorithm PREDICT-AND-DOUBLE (Algorithm 2). Here, we use a hyper-parameter $\epsilon$ that offers a tradeoff between the consistency and robustness of the algorithm. We also use the following definition:

**Definition 3.** *Given an input length* $\tau$ *and any* $\alpha > 0$*, we use* MAX-LENGTH$(\alpha, \tau)$ *to denote the largest length* $t$ *such that* OPT$(t) \leq \alpha \cdot$ OPT$(\tau)$*.*

As described in the introduction, the desiderata for an online algorithm with predictions are its consistency and robustness; we establish the tradeoff between these parameters for the PREDICT-AND-DOUBLE algorithm below.

---
**Algorithm 2** PREDICT-AND-DOUBLE
---

**Input:** The input sequence $\mathcal{I}$ and prediction $\hat{T}$.
**Output:** The online solution SOL.

Set SOL $:= \emptyset$, $t_1 := $ MIN-LENGTH$(\epsilon/5, \hat{T})$, and $t_2 := $ MAX-LENGTH$(1 + \epsilon/5, \hat{T})$.
**Phase 1:** Execute DOUBLE while $t < t_1$.
**Phase 2:** At $t = t_1$, add OPT$(t_2)$ to SOL.
**Phase 3:** If $t > t_2$, resume DOUBLE as follows:
Set $i := 0$, $\tau_0 := t_2 + 1$.
**for** $t = t_2 + 1, t_2 + 2, \ldots, T$
   **if** $t = \tau_i$
      Set $\tau_{i+1} = $ MIN-LENGTH$(2, \tau_i)$.
      Add OPT$(\tau_{i+1} - 1)$ to SOL.
      Increment $i$.
---

**Theorem 2.** *The* PREDICT-AND-DOUBLE *algorithm has a consistency of* $1 + \epsilon$ *and robustness of* $5\left(1 + \frac{1}{\epsilon}\right)$.

We also show that this tradeoff between $(1 + \epsilon)$-consistency and $O(1/\epsilon)$-robustness bounds is essentially tight.

**Theorem 3.** *Any algorithm for the* ONLINESEARCH *problem with predictions that has a consistency bound of* $1 + \epsilon$ *must have a robustness bound of* $\Omega\left(\frac{1}{\epsilon}\right)$.

Having shown the consistency and robustness of the PREDICT-AND-DOUBLE algorithm, we now analyze how its competitive ratio varies with error in the prediction $\hat{T}$. In particular, the next lemma shows that the competitive ratio gracefully degrades with prediction error for small error, and is capped at 4 for large error.

**Lemma 4.** *Given a prediction* $\hat{T}$ *for the input length, the competitive ratio of* PREDICT-AND-DOUBLE *is given by:*

$$\text{CR} \leq \begin{cases} 4 \, , T \leq t_1 \\ (1 + \epsilon) \cdot \frac{\text{OPT}(\hat{T})}{\text{OPT}(T)} \, , t_1 \leq T \leq t_2 \\ 4 \, , T > t_2 \end{cases}$$

*where* $t_1$ *represents the minimum value of* $t$ *that satisfies* OPT$(t) \geq \frac{\epsilon}{5} \cdot$ OPT$(\hat{T})$ *and* $t_2$ *represents the maximum value of* $t$ *that satisfies* OPT$(t) \leq (1 + \frac{\epsilon}{5}) \cdot$ OPT$(\hat{T})$.

## 4 LEARN TO SEARCH: A Regression Approach

In the previous section, we designed an algorithm for the ONLINESEARCH problem that utilizes ML predictions. Now, we delve deeper into how we can generate these predictions. More generally, we develop a regression-based approach to learn to solve an ONLINESEARCH problem. For this purpose, we first introduce some standard terminology for our learning framework, which we call the LEARNTOSEARCH problem.

### 4.1 Preliminaries

An instance $(x, z)$ of the LEARNTOSEARCH problem is given by a feature $x \in \mathbb{X}$, and the (unknown) cost of the optimal offline solution $z \in [1, M]$. The two quantities $x$ and $z$ are assumed to be drawn from a joint distribution on $\mathbb{X} \times [1, M]$. A prediction strategy works with a hypothesis class $\mathcal{F}$ that is a subset of functions $\mathbb{X} \mapsto [1, M]$ and tries to obtain the best function $f \in \mathcal{F}$ that predicts the target variable $z$ accurately. For notational convenience, we set our target $y = \ln z$, i.e., we try to predict the log-cost of the optimal solution. Note that predicting the log-cost of OPT$(T)$ is equivalent to predicting the input length $T$.[2] Furthermore, let $\mathbb{D}$ denote the input distribution on $\mathbb{X} \times \mathbb{Y}$, where $\mathbb{Y} = [0, H]$ and $H = \ln M$; i.e., we assume that $(x, y) \sim \mathbb{D}$.

---
[2]When multiple input lengths might have the same optimal cost, we can just pick the longest one.

We define a LEARNTOSEARCH algorithm $\mathcal{A}$ as a strategy that receives a set of $m$ samples $S \sim \mathbb{D}^m$ for *training*, and later, when given the feature set $x$ of a *test* instance $(x, y) \sim \mathbb{D}$ (where $y$ is not revealed to the algorithm), it defines an online algorithm for the ONLINESEARCH problem with input $y$. Recall that an online algorithm constitutes a sequence of solutions that the algorithm buys at different times of the input sequence (see Algorithm 3 for a generic description of an LEARNTOSEARCH algorithm).

---

**Algorithm 3** A general LEARNTOSEARCH algorithm

---

**Training:** Given a Sample Set $S$, the training phase outputs a mapping $M$ from every feature vector $x \in \mathbb{X}$ to an increasing sequence of positive integers
**Testing:** Given unknown sample $x \in \mathbb{X}$, define thresholds $M(x) = (\tau_0, \tau_1 \ldots)$
Set $i := 0, \text{SOL} := \text{OPT}(\tau_0 - 1)$.
**while** (Input has not ended)
   **if** (SOL is infeasible)
      $\text{SOL} := \text{OPT}(\tau_{i+1} - 1)$.
      Increment $i$.

---

We will use the notation $\text{CR}_{\mathcal{A}}(x, y)$ to denote the competitive ratio obtained by an algorithm $\mathcal{A}$ on the instance $(x, y)$. For a given set of thresholds $(\tau_0, \tau_1 \ldots)$, define $i_T = \min_{\tau_i > T} i$. Then, $\mathcal{A}$ pays a total cost of $\sum_{i=0}^{i_T} \text{OPT}(\tau_i)$, and thus the competitive ratio is

$$\text{CR}_{\mathcal{A}}(x, y) = \frac{\sum_{i=0}^{i_T} \text{OPT}(\tau_i)}{e^y}.$$

We define the "efficiency" of a LEARNTOSEARCH algorithm by comparing its performance with the best achievable competitive ratio. The optimal competitive ratio for a given distribution may be strictly greater than 1. For example, consider the distribution where $x$ is fixed (say $x_0$) and $z$ is uniformly distributed over the set $\{2, 4\}$. One can verify that the best strategy for the above distribution is to buy the solution costing 2, and then if the input has not ended, then buy the solution costing 4. The competitive ratio for this strategy (in expectation) is 1.25.

**Definition 4.** *A* LEARNTOSEARCH *algorithm $\mathcal{A}$ is said to be $\epsilon$-efficient if*

$$\mathbb{E}_{(x,y)\sim\mathbb{D}}\text{CR}_{\mathcal{A}}(x, y) \leq \rho^* + \epsilon,$$

*where $\rho^* = \mathbb{E}_{(x,y)\sim\mathbb{D}}\text{CR}_{\mathcal{A}^*}(x, y)$ and $\mathcal{A}^*$ is an optimal solution that has full knowledge of $\mathbb{D}$ and no computational limitations.*

The "expressiveness" of a function family is captured by the following standard definition:

**Definition 5.** *A set $S = \{x_1, x_2, \ldots x_m\}$ is said to be "shattered" by a class $\mathcal{F}$ of real-valued functions $S \mapsto [0, H]$ if there exists "witnesses" $R = \{r_1, r_2 \ldots r_m\} \in [0, H]^m$ such that the following condition holds: For all subsets $T \subseteq S$, there exists an $f \in \mathcal{F}$ such that $f(x_i) > r_i$ if and only if $x_i \in T$. The "pseudo-dimension" of $\mathcal{F}$ (denoted as $Pdim(\mathcal{F})$) is the cardinality of the largest subset $S \subseteq X$ that is shattered by $\mathcal{F}$.*

### 4.2 The Sample Complexity of LEARNTOSEARCH

Our overall strategy is to learn a suitable predictor function $f \in \mathcal{F}$ and use $f(x)$ as a prediction in the PREDICT-AND-DOUBLE algorithm. Note that prediction errors on the two sides (over- and under-estimation) affect the competitive ratio of PREDICT-AND-DOUBLE (given by Lemma 4) in different ways. If we underestimate $\text{OPT}(T)$ by a factor less than $1 + \frac{\epsilon}{5}$, the competitive ratio remains $1 + O(\epsilon)$, but a larger underestimate causes the competitive ratio to climb up to 4. On the other hand, if we overestimate $\text{OPT}(T)$, then the competitive ratio grows steadily by the ratio of over-estimation, until it reaches $5 \cdot \left(1 + \frac{1}{\epsilon}\right)$. This asymmetric dependence is illustrated in Figure 1.

At a high level, our goal is to use regression to obtain the best function $f \in \mathcal{F}$. But, the asymmetric behavior of the competitive ratio suggests that we should not use a standard loss function in the regression analysis. Let $\epsilon$ be the accuracy parameter for the PREDICT-AND-DOUBLE algorithm, and let $\hat{y} = \ln \text{OPT}(\hat{T})$ and $y = \ln \text{OPT}(T)$ be the predicted and actual log-cost of the optimal solution respectively. Then we define the following loss function that follows the asymmetric behaviour of the competitive ratio for PREDICT-AND-DOUBLE:

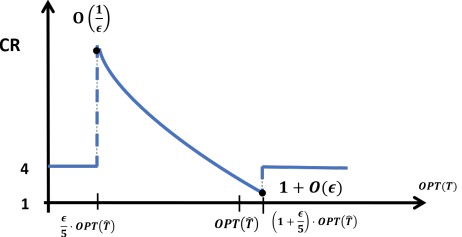

Figure 1: Competitive ratio of the PREDICT-AND-DOUBLE algorithm for a fixed prediction $\hat{T}$ as a function of the input $T$, where the prediction is $\hat{T}$

**Definition 6.** *The $\epsilon$-parameterized competitive error is defined as:*

$$\ell_\epsilon(y, \hat{y}) = \begin{cases} \frac{5}{\epsilon} - 1 \text{ when } y \leq \hat{y} - \ln \frac{5}{\epsilon} \\ e^{y-\hat{y}} - 1 \text{ when } \hat{y} - \ln \frac{5}{\epsilon} < y \leq \hat{y} \\ \frac{1}{\epsilon} \cdot (y - \hat{y}) \text{ when } \hat{y} < y \leq \hat{y} + \ln \left(1 + \frac{\epsilon}{5}\right) \\ 1 \text{ when } y > \hat{y} + \ln \left(1 + \frac{\epsilon}{5}\right). \end{cases}$$

We give more justification for using this loss function in the supplementary material, and show that standard loss functions do not suffice for our purposes. Using this loss function, we can measure the error of a function for an input distribution or for a fixed input set:

**Definition 7.** *Given a distribution $\mathbb{D}$ on the set $\mathbb{X} \times \mathbb{Y}$ and function $f : \mathbb{X} \mapsto \mathbb{Y}$, we define*

$$\mathbf{ER}_{\mathbb{D},\epsilon}(f) = \mathbb{E}_{(x,y)\sim\mathbb{D}}[\ell_\epsilon(y, f(x))].$$

*Alternatively, for a set of samples, $S \sim \mathbb{D}^m$, we define,*

$$\mathbf{ER}_{S,\epsilon}(f) = \frac{1}{m} \cdot \sum_{i=1}^{m} \ell_\epsilon(y_i, f(x_i)).$$

Our high-level goal is to use samples to optimize for the loss function called $\epsilon$-parameterized competitive error that we defined above over the function class $\mathcal{F}$, and then use an algorithm that translates the empirical error bound to a competitive ratio bound. This requires, in the training phase, that we optimize the empirical loss on the training samples. We define such a minimizer below:

**Definition 8.** *For a given set of samples $S \sim \mathbb{D}$ and a function family $\mathcal{F}$, we denote an optimization scheme $\mathcal{O} : S \mapsto \mathcal{F}$ as $\epsilon-$Sample Error Minimizing (SEM) if it returns a function $\hat{f} \in \mathcal{F}$ satisfying:*

$$\mathbf{ER}_{S,\epsilon}(\hat{f}) \leq \inf_{f\in\mathcal{F}} \left[\mathbf{ER}_{S,\epsilon}(f)\right] + \epsilon.$$

For the rest of this paper, we will assume that we are given an $\epsilon-$ SEM routine for arbitrary $\epsilon > 0$.

We are now ready to present our LEARNTOSEARCH algorithm (Algorithm 4), which basically uses the predictor with minimum expected loss to make predictions for PREDICT-AND-DOUBLE.

---

**Algorithm 4** A LEARNTOSEARCH algorithm with accuracy parameter $\epsilon$

---

**Training:**
    **Input:** Sample Set $S$, Function Family $\mathcal{F}$
    **Output:** $\hat{f}$ output by an $\epsilon$-SEM algorithm $\mathcal{O}$, i.e., $\mathbf{ER}_{S,\epsilon}(\hat{f}) \leq \inf_{\tilde{f}\in\mathcal{F}} \mathbf{ER}_{S,\epsilon}(\tilde{f}) + \epsilon$.
**Testing:**
    Given new sample $x$, set $\hat{y} = \hat{f}(x)$.
    Predicted prefix length: $\hat{T} = \max_{\text{OPT}(t)\leq e^{\hat{y}}} t$.
    Call PREDICT-AND-DOUBLE with $\hat{T}$ and $\epsilon$.

---

We relate the competitive ratio of Algorithm 4 to the error of function $\hat{f}$ obtained during training:

**Lemma 5.** *Algorithm 4 has a competitive ratio upper bounded by $\left(1 + \epsilon + 3 \cdot \mathbf{ER}_{\mathbb{D},\epsilon}(\hat{f})\right)$.*

**Standard and Agnostic Models.** We consider two different settings. First, we assume that the function class $\mathcal{F}$ contains the function $f^*$ that maps the feature set $x$ to $y$ – we call this the *standard* model. We relax this assumption in the more general *agnostic* model, where the function class $\mathcal{F}$ is arbitrary. In terms of the error function, in the standard model, we have $\inf_{f \in \mathcal{F}} \mathbf{ER}_{\mathbb{D},\epsilon}(f) = \inf_{f \in \mathcal{F}} \mathbf{ER}_{S,\epsilon}(f) = 0$, while no such guarantee holds in the agnostic model.

## 4.3 Analysis in the Standard Model

Next, we analyze the competitive ratio of Algorithm 4 in the standard model, i.e., when $\inf_{f \in \mathcal{F}} \mathbf{ER}_{\mathbb{D},\epsilon}(f) = \inf_{f \in \mathcal{F}} \mathbf{ER}_{S,\epsilon}(f) = 0$.

**Theorem 6.** *In the standard model, Algorithm 4 obtains a competitive ratio of $1 + O(\epsilon)$ with probability at least $1 - \delta$, when using $O\left(\frac{H \cdot d \log \frac{1}{\epsilon} \log \frac{1}{\delta}}{\epsilon}\right)$ samples, where $d = Pdim(\mathcal{F})$.*

When the cost of the optimal solution $\text{OPT}(\tau)$ is hard to compute, we can replace the offline optimal with an online algorithm that achieves competitive ratio $c$ given the value of $\tau$ to get the following:

**Corollary 7.** *In the standard model, if there exists a $c$-competitive algorithm for $\text{OPT}(\tau)$ given the value of prefix-length $\tau$, Algorithm 4 obtains a competitive ratio of $c(1 + O(\epsilon))$ with probability at least $1 - \delta$, when using $O\left(\frac{H \cdot d \log \frac{1}{\epsilon} \log \frac{1}{\delta}}{\epsilon}\right)$ samples, where $d = Pdim(\mathcal{F})$.*

We also show that the result in Theorem 6 is tight up to a factor of $H \log 1/\epsilon$:

**Theorem 8.** *Let $\mathcal{F}$ be a family of real valued functions such that there exists a function $f^* : \mathbb{X} \mapsto \mathbb{Y}$ that $f^*(x) = y$ and let $d = Pdim(\mathcal{F})$. There exists an instance of the LEARNTOSEARCH problem that enforces any algorithm to query $\Omega\left(\frac{d \log \frac{1}{\delta}}{\epsilon}\right)$ samples in order to have an expected competitive ratio of $1 + \epsilon$ with probability $\geq 1 - \delta$.*

## 4.4 Extension to the Agnostic Model

In the agnostic model, we no longer assume a function $f \in \mathcal{F}$ that predicts the log-cost $y$ perfectly. It is possible that the true predictor is outside $\mathcal{F}$, or in more difficult scenarios for any feature $x$, the behaviour of the log-cost $y$ may be entirely arbitrary.

We first show that the loss function $\epsilon$-parameterized competitive error defined earlier is still a reasonable proxy for the competitive ratio. Specifically, we show that any algorithm that hopes to achieve a competitive ratio of $1 + O(\epsilon)$ must use a prediction $\hat{f} \in \mathcal{F}$ whose error $\mathbf{ER}_{\mathbb{D},\epsilon}(f)$ is bounded by $O(\epsilon)$. We formally state this below:

**Lemma 9.** *Let $\mathcal{A}$ be an algorithm for LEARNTOSEARCH that has access to a predictor $\hat{f} : \mathbb{X} \mapsto [0, H]$ for the log-cost $y$. Then, there exists a distribution $\mathbb{D}$ and a function $\hat{f}_{\mathcal{A}}$ with the property $\mathbf{ER}_{\mathbb{D},\epsilon}(\hat{f}_{\mathcal{A}}) = \epsilon$ such that $\mathbb{E}_{(x,y) \sim \mathbb{D}}[\text{CR}_{\mathcal{A}}(x, y)] \geq 1 + \frac{\epsilon}{2}$.*

Unlike in the standard model, we no longer have that for any $\epsilon > 0$, $\min_{f \in \mathcal{F}} \mathbf{ER}_{\mathbb{D},\epsilon}(f) = 0$. Therefore, we need to first quantify the performance of an *ideal* algorithm that uses predictors from $\mathcal{F}$.

**Definition 9.** *Let $\chi(\epsilon) = \min_{f \in \mathcal{F}} \mathbf{ER}_{\mathbb{D},\epsilon}(f)$. Then, $\Delta_{\mathcal{F}}$ is the solution to the equation: $\epsilon = \chi(\epsilon)$.*

$\Delta_{\mathcal{F}}$ measures the best competitive ratio that we can hope to get when we use a predictor from $\mathcal{F}$. Note that $\epsilon$ appears in two places in this definition, since the loss function in Definition 6 depends on $\epsilon$. We first show that this is a reasonable definition in that the solution to the equation is unique:

**Lemma 10.** *For a given function family $\mathcal{F}$ and distribution $\mathbb{D}$, the value of $\Delta_{\mathcal{F}}$ is unique.*

We also give an algorithm that can approximate $\Delta_{\mathcal{F}}$ (Algorithm 5).

**Lemma 11.** *If $|S| \geq C \cdot \left(\frac{H \cdot d \log \frac{1}{\epsilon} \log \frac{1}{\delta}}{\epsilon}\right)$ for suitable constants $C > 0, \delta \leq \frac{1}{2}$, and $\epsilon \leq \Delta_{\mathcal{F}}$, then with probability at least $1 - \delta$, we have $\varepsilon/6 \leq \Delta_F \leq 5\varepsilon/3$, where $\varepsilon$ is as returned by Algorithm 5.*

We are now ready to define our LEARNTOSEARCH algorithm for the agnostic model. This algorithm is simply Algorithm 4 where the accuracy parameter $\epsilon$ is set to the value of $\varepsilon$ returned by Algorithm 5.

---

**Algorithm 5** Procedure to estimate $\Delta_{\mathcal{F}}$

---

**Input:** Sample Set $S$, and function family $\mathcal{F}$

Let $\epsilon$ be an accuracy parameter given by the size of the sample set $S$.

Choose $\varepsilon := \epsilon$

Compute: $\hat{f}$ such that $\mathbf{ER}_{S,\varepsilon}(\hat{f}) \leq \min_{f \in \mathcal{F}} \mathbf{ER}_{S,\varepsilon}(f) + \frac{\varepsilon}{3}$.

**while** $\varepsilon \leq \mathbf{ER}_{S,\varepsilon}(\hat{f})$

    $\varepsilon \leftarrow 2\varepsilon$.

    Recompute $\hat{f}$ s.t. $\mathbf{ER}_{S,\varepsilon}(\hat{f}) \leq \min_{f \in \mathcal{F}} \mathbf{ER}_{S,\varepsilon}(f) + \frac{\varepsilon}{3}$.

**Return** $\varepsilon$.

---

**Theorem 12.** *In the agnostic model for a function family $\mathcal{F}$, Algorithm 4 with accuracy parameter $\varepsilon$ from Algorithm 5 obtains a competitive ratio of $1 + O\left(\Delta_{\mathcal{F}}\right)$ with probability at least $1 - \delta$, when using $O\left(\frac{H \cdot d \log\left(\frac{1}{\Delta_{\mathcal{F}}}\right) \cdot \log \frac{1}{\delta}}{\Delta_{\mathcal{F}}}\right)$ samples, where $d = Pdim(\mathcal{F})$.*

We also lower bound the sample complexity of a LEARNTOSEARCH algorithm:

**Theorem 13.** *Any LEARNTOSEARCH algorithm that is $\epsilon$-efficient with probability at least $1 - \delta$ must query $\Omega\left(\frac{\log \frac{1}{\delta}}{\epsilon^2}\right)$ samples.*

### 4.5 Robustness of Algorithm 4

So far, we have established the competitive ratio of Algorithm 4 in the PAC model. Now, we show the robustness of this algorithm, i.e., bound its competitive ratio for *any* input. Even for adversarial inputs, we show that this algorithm has a competitive ratio of $O(1/\epsilon)$, which matches the robustness guarantees in Theorem 2 for the PREDICT-AND-DOUBLE algorithm.

**Theorem 14.** *Algorithm 4 is $5(1 + \frac{1}{\epsilon}) = O\left(\frac{1}{\epsilon}\right)$-robust.*

## 5 Conclusion, Limitations, and Future Work

In this paper, we studied the role of regression in making predictions for learning-augmented online algorithms. In particular, we used the ONLINESEARCH framework that includes a variety of online problems such as ski rental and its generalizations, online scheduling, online bin packing, etc. and showed that by using a carefully crafted loss function, we can obtain predictions that yield near-optimal algorithms for this problem. One assumption that holds for the above problems, but not for other problems such as online matching, is the composability of solutions, i.e., that the union of two feasible solutions is also a feasible solution. Extending our work to such "packing" problems is an interesting direction for future research. Another interesting direction would be to give a general recipe for converting competitive ratios to loss functions, minimizing which over a collection of training samples generates better ML predictions for online problems.

## 6 Acknowledgements

This research was partially funded by the Indo-US Virtual Networked Joint Center project No. IUSSTF/JC-017/2017. Keerti Anand and Debmalya Panigrahi were supported in part by NSF Awards CCF-1955703, CCF-1750140 (CAREER), and ARO Award W911NF2110230. Rong Ge was also supported in part by NSF Awards DMS-2031849, CCF-1704656, CCF-1845171 (CAREER), CCF-1934964 (TRIPODS), a Sloan Research Fellowship, and a Google Faculty Research Award.

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
