# A Proofs for the Upper Bounds (Theorems 6 and 12)

In this section, we prove Theorems 6 and 12 which give upper bounds on the sample complexity in the standard and agnostic settings respectively. In order to have sample complexity bounds relating to the pseudo dimension of the function class, we would need to introduce the notion of covering numbers and relate them to the pseudo-dimension.

**Definition 10.** *Given a set $S$ in Euclidean space and a metric $d(\cdot, \cdot)$, the set $W \subseteq S$ is said to be $\epsilon$ cover of $S$ if for any $s \in S$, there exists a $w \in W$ such that $d(s, w) \leq \epsilon$. The smallest possible cardinality of such an $\epsilon$ cover is known as the $\epsilon$ covering number of $S$ with respect to $d$ and is denoted as $\mathcal{N}_{d(\cdot, \cdot)}(\epsilon, S)$.*

When $d$ is given by the distance metric

$$d_p(r, s) = \Big| \sum_{i=1}^{d} (r_i - s_i)^p \Big|^{1/p},$$

where $r = (r_1, r_2 \ldots r_d), s = (s_1, s_2 \ldots s_d) \in \mathbb{R}^d$, we will denote the $\epsilon$ covering number of set $S$ as $\mathcal{N}_p(\epsilon, S)$. For a given real-valued function family $\mathcal{F}$ and $x = (x_1, x_2, \ldots, x_m) \in \mathbb{X}^m$, we denote

$$\mathcal{F}_{|x} = \{(f(x_1), f(x_2), \ldots f(x_m)) \mid f \in \mathcal{F}\} \quad \text{and}$$

$$\mathcal{N}_p(\epsilon, \mathcal{F}, m) = \sup_{x \in \mathbb{X}^m} \big[ \mathcal{N}_p(\epsilon, \mathcal{F}_{|x}) \big].$$

Note that $\mathcal{N}_1(\epsilon, \mathcal{F}, m) \leq \mathcal{N}_2(\epsilon, \mathcal{F}, m) \leq \mathcal{N}_\infty(\epsilon, \mathcal{F}, m)$.

The following is a well-known result that relates covering numbers to the pseudo dimension (cf. Theorem 12.2 in Book [8]):

**Lemma 15.** *Let $\mathcal{F}$ be a real-valued function family with pseudo dimension $d$, then for any $\epsilon \leq \frac{1}{d}$, we have*

$$\mathcal{N}_1(\epsilon, \mathcal{F}, m) \leq O\left(\frac{1}{\epsilon^d}\right).$$

## A.1 The Standard Model: Proof of Theorem 6

First, we relate covering numbers to this error measure. This will be crucial in proving Theorem 6.

**Lemma 16.** *Let $\mathbb{D}$ be a distribution on $\mathbb{X} \times \mathbb{Y}$ and let $S \in \mathbb{D}^m$. For $0 \leq \eta \leq 12$ and $m \geq \frac{8 \cdot H}{\epsilon \eta^2}$, for any real valued function family $\mathcal{F}$, we have:*

$$\mathbb{P}_{S \in \mathbb{D}^m} \left[ \sup_{f \in \mathcal{F}} \frac{\big| \mathbf{ER}_{S, \epsilon}(f) - \mathbf{ER}_{\mathbb{D}, \epsilon}(f) \big|}{\mathbf{ER}_{\mathbb{D}, \epsilon}(f) + \epsilon} \geq \eta \right] \leq 4 \cdot \mathcal{N}_1\left(\frac{\eta \epsilon^2}{8}, \mathcal{F}, 2m\right) \cdot \exp\left(-\frac{m \cdot \eta^2 \cdot}{64 H}\right).$$

To prove this lemma, we need the following definition.

**Definition 11.** *The $\nu$-normalised error is defined as :*

$$\hat{\mathbf{ER}}_{S, \mathbb{D}, \nu}(f) = \frac{\mathbf{ER}_{S, \nu}(f)}{\mathbf{ER}_{D, \nu}(f) + \nu}.$$

Note that $\nu$ plays two roles here: as a parameter in the loss function, and as the normalization parameter.

We will drop the $\mathbb{D}, \epsilon$ in the subscript and simply refer to $\hat{\mathbf{ER}}_{S, \mathbb{D}, \epsilon}(f)$ as $\hat{\mathbf{ER}}_S(f)$ for the rest of the proof. Additionally, we denote $\hat{\mathbf{ER}}_{\mathbb{D}}(f) = \frac{\mathbf{ER}_{\mathbb{D}}(f)}{\mathbf{ER}_{\mathbb{D}}(f) + \epsilon}$.

We break this proof into four separate claims as illustrated below.

First, we reduce the probability of the event: $\big[ \big| \hat{\mathbf{ER}}_S(f) - \hat{\mathbf{ER}}_{\mathbb{D}}(f) \big| \geq \eta \big]$ to a probability term involving two sample sets $S, \bar{S}$ the members of which are drawn i.i.d.

**Lemma 17.** *For $m \geq \frac{8H}{\epsilon \cdot \eta^2}$, we have that:*

$$\mathbb{P}_{S \sim \mathbb{D}^m} \left[ \big| \hat{\mathbf{ER}}_S(f) - \hat{\mathbf{ER}}_{\mathbb{D}}(f) \big| \geq \eta \right] \leq 2 \cdot \mathbb{P}_{(S, \bar{S}) \sim D^m \times D^m} \left[ \big| \hat{\mathbf{ER}}_S(f) - \hat{\mathbf{ER}}_{\bar{S}}(f) \big| \geq \eta/2 \right].$$

*Proof.* For a given sample $S \sim \mathbb{D}^m$, let $f_{bad}^S \in \mathcal{F}$ denote a function such that $\left| \hat{\mathbf{ER}}_S(f) - \hat{\mathbf{ER}}_\mathbb{D}(f) \right| \geq \eta$ if it exists and any fixed function in the family otherwise.

$$
\begin{aligned}
\mathbb{P}_{(S,\bar{S})\sim D^m \times D^m} & \left[ \sup_{f \in \mathcal{F}} \left| \hat{\mathbf{ER}}_S(f) - \hat{\mathbf{ER}}_{\bar{S}}(f) \right| \geq \frac{\eta}{2} \right] \\
& \geq \mathbb{P}_{(S,\bar{S})\sim D^m \times D^m} \left[ \left| \hat{\mathbf{ER}}_S(f_{bad}^S) - \hat{\mathbf{ER}}_{\bar{S}}(f_{bad}^S) \right| \geq \frac{\eta}{2} \right] \\
& \geq \mathbb{P}_{(S,\bar{S})\sim D^m \times D^m} \left[ \left| \hat{\mathbf{ER}}_S(f_{bad}^S) - \hat{\mathbf{ER}}_\mathbb{D}(f_{bad}^S) \right| \geq \eta \cap \left| \hat{\mathbf{ER}}_S(f_{bad}^S) - \hat{\mathbf{ER}}_{\bar{S}}(f_{bad}^S) \right| \leq \frac{\eta}{2} \right] \\
& = \mathbb{P}_{S\sim \mathbb{D}^m} \left[ \left| \hat{\mathbf{ER}}_S(f_{bad}^S) - \hat{\mathbf{ER}}_\mathbb{D}(f_{bad}^S) \right| \geq \eta \right] \cdot \mathbb{P}_{\bar{S}\sim \mathbb{D}^m | S} \left[ \left| \hat{\mathbf{ER}}_S(f_{bad}^S) - \hat{\mathbf{ER}}_{\bar{S}}(f_{bad}^S) \right| \leq \frac{\eta}{2} \right] \\
& = \mathbb{P}_{S\sim \mathbb{D}^m} \left[ \sup_{f \in \mathcal{F}} \left| \hat{\mathbf{ER}}_S(f) - \hat{\mathbf{ER}}_\mathbb{D}(f) \right| \geq \eta \right] \cdot \mathbb{P}_{\bar{S}\sim \mathbb{D}^m | S} \left[ \left| \hat{\mathbf{ER}}_S(f_{bad}^S) - \hat{\mathbf{ER}}_{\bar{S}}(f_{bad}^S) \right| \leq \frac{\eta}{2} \right].
\end{aligned}
$$

Now, the term $\mathbb{P}_{\bar{S}\sim \mathbb{D}^m | S} \left[ \left| \hat{\mathbf{ER}}_S(f_{bad}^S) - \hat{\mathbf{ER}}_{\bar{S}}(f_{bad}^S) \right| \leq \frac{\eta}{2} \right]$ is bounded below by the Chebyshev's inequality as follows:

$$
\mathbb{P}_{\bar{S}\sim \mathbb{D}^m | S} \left[ \left| \hat{\mathbf{ER}}_S(f_{bad}^S) - \hat{\mathbf{ER}}_{\bar{S}}(f_{bad}^S) \right| \geq \frac{\eta}{2} \right] \geq 1 - \frac{\mathrm{Var}_{\bar{S}\sim \mathbb{D}^m | S} \left[ \hat{\mathbf{ER}}_{\bar{S}}(f_{bad}^S) \right]}{\frac{\eta^2}{4}} \geq 1 - \left( \frac{H}{m \cdot \epsilon \cdot \frac{\eta^2}{4}} \right) \geq \frac{1}{2}.
$$

Note that the random variable $\hat{\mathbf{ER}}_{\bar{S}}(f_{bad}^S)$ is $\frac{1}{m}$ times the sum of $m$ i.i.d. random variables each having variance bounded above by $\frac{H}{\epsilon}$. Hence, $\mathrm{Var}_{\bar{S}\sim \mathbb{D}^m | S} \left[ \hat{\mathbf{ER}}_{\bar{S}}(f_{bad}^S) \right] \leq \frac{1}{m} \cdot \frac{H}{\epsilon}$, and the second inequality above follows. The last inequality is immediate from the fact that $m \geq \left( \frac{8 \cdot H}{\epsilon \eta^2} \right)$, which proves our claim. $\square$

The second claim intuitively says that the probabilities remain unchanged under symmetric permutations. Let $\sigma$ denote a permutation on the set $\{1, 2, \ldots, 2m\}$ such that for each $i \in \{1, 2, \ldots, m\}$, we use either of the two mappings:

- $\sigma(i) = i$ and $\sigma(m + i) = m + i$, or
- $\sigma(i) = m + i$ and $\sigma(m + i) = i$.

Let $\Gamma^m$ denote the set of all such permutations $\sigma$. Suppose we draw i.i.d. samples $S \sim \mathbb{D}^m$ and $\bar{S} \sim \mathbb{D}^m$; let $S = \{s_1, s_2, \ldots, s_m\}$ and $\bar{S} = \{\bar{s}_1, \bar{s}_2, \ldots, \bar{s}_m\}$. Then, define $\sigma(S)$ and $\sigma(\bar{S})$ by using a permutation $\sigma \in \Gamma^m$ as follows. Let $\sigma(S) = \{s_1', s_2', \ldots, s_m'\}$ and $\sigma(\bar{S}) = \{\bar{s}_1', \bar{s}_2', \ldots, \bar{s}_m'\}$ such that $s_i' = s_i$ and $\bar{s}_i' = \bar{s}_i$ if $\sigma(i) = i$ and $\sigma(m + i) = m + i$, while $s_i' = \bar{s}_i$ and $\bar{s}_i' = s_i$ if $\sigma(i) = m + i$ and $\sigma(m + i) = i$. Let $U^m$ denote the uniform distribution over $\Gamma^m$.

**Lemma 18.** *For every $f \in \mathcal{F}$:*

$$
\mathbb{P}_{(S,\bar{S})\sim D^m \times D^m} \left[ \left| \hat{\mathbf{ER}}_S(f) - \hat{\mathbf{ER}}_{\bar{S}}(f) \right| \geq \eta/2 \right] \leq \sup_{(S,\bar{S})\in (\mathbb{X}\times\mathbb{Y})^{2m}} \left( \mathbb{P}_{\sigma\sim U^m} \left[ \left| \hat{\mathbf{ER}}_{\sigma(S)}(f) - \hat{\mathbf{ER}}_{\sigma(\bar{S})}(f) \right| \geq \eta/2 \right] \right).
$$

*Proof.* We have for every $f \in \mathcal{F}$:

$$
\begin{aligned}
\mathbb{P}_{(S,\bar{S})\sim D^m \times D^m} & \left[ \left| \hat{\mathbf{ER}}_S(f) - \hat{\mathbf{ER}}_{\bar{S}}(f) \right| \geq \eta/2 \right] \\
& = \mathbb{P}_{(S,\bar{S})\sim D^m \times D^m, \, \sigma\sim U^m} \left[ \left| \hat{\mathbf{ER}}_{\sigma(S)}(f) - \hat{\mathbf{ER}}_{\sigma(\bar{S})}(f) \right| \geq \eta/2 \right] \quad \text{(by the i.i.d. property)} \\
& \leq \sup_{(S,\bar{S})\in (\mathbb{X}\times\mathbb{Y})^{2m}} \left( \mathbb{P}_{\sigma\sim U^m} \left[ \left| \hat{\mathbf{ER}}_{\sigma(S)}(f) - \hat{\mathbf{ER}}_{\sigma(\bar{S})}(f) \right| \geq \eta/2 \right] \right),
\end{aligned}
$$

where in the last expression we chose the members of $S, \bar{S}$ adversarially instead of randomly. $\square$

Third, we make use of covering numbers to quantify the above probability.

**Lemma 19.** *Fix a $(S, \bar{S}) \in (\mathbb{X} \times \mathbb{Y})^{2m}$. Consider the set $\mathcal{G} \in \mathcal{F}$ such that $\hat{\ell}_{\mathcal{G}}(S, \bar{S})$ is a $\frac{\eta}{8}$-covering of the set $\hat{\ell}_{\mathcal{F}}(S, \bar{S}) = \{\hat{\ell}_f(x_i, y_i) \mid (x_i, y_i) \in S \cup \bar{S}, f \in \mathcal{F}\} \in [0, H]^{2m}$ (wrt $d_1(\cdot, \cdot)$). Then :*

$$\mathbb{P}_{S \sim \mathbb{D}^m} \left[ \sup_{f \in \mathcal{F}} \left| \hat{\mathbf{ER}}_S(f) - \hat{\mathbf{ER}}_{\mathbb{D}}(f) \right| \geq \eta \right] \leq \mathcal{N}\left(\frac{\eta}{8}, \hat{\ell}_f, 2m\right) \cdot \max_{g \in \mathcal{G}} \mathbb{P}_{S \sim \mathbb{D}^m} \left[ \left| \hat{\mathbf{ER}}_{\sigma(S)}(g) - \hat{\mathbf{ER}}_{\sigma(\bar{S})}(g) \right| \geq \frac{\eta}{4} \right].$$

*Proof.* Note that the cardinality of $\mathcal{G}$ is less than $\mathcal{N}_1(\eta/8, \ell_F, 2m)$ and is a bounded number. We claim that whenever an $f \in \mathcal{F}$ satisfies, $\left| \hat{\mathbf{ER}}_{\sigma(S)}(f) - \hat{\mathbf{ER}}_{\sigma(\bar{S})}(f) \right| \geq \frac{\eta}{2}$, then there exists a $g \in \mathcal{G}$ such that, $\left| \hat{\mathbf{ER}}_{\sigma(S)}(g) - \hat{\mathbf{ER}}_{\sigma(\bar{S})}(g) \right| \geq \frac{\eta}{4}$.

Let $g$ satisfy that, $\frac{1}{2m}[\sum_{i=1}^{2m} |\ell_g(x_i, y_i) - \ell_g(x_i, y_i)|] \leq \frac{\eta}{8}$.

We are guaranteed that such a $g$ exists, since it is in the cover.

$$\frac{\eta}{2} \leq \left| \hat{\mathbf{ER}}_{\sigma(S)}(f) - \hat{\mathbf{ER}}_{\sigma(\bar{S})}(f) \right|$$

$$= \left| (\hat{\mathbf{ER}}_{\sigma(S)}(f) - \hat{\mathbf{ER}}_{\sigma(S)}(g)) - (\hat{\mathbf{ER}}_{\sigma(\bar{S})}(f) - \hat{\mathbf{ER}}_{\sigma(\bar{S})}(g)) + (\hat{\mathbf{ER}}_{\sigma(S)}(g) - \hat{\mathbf{ER}}_{\sigma(\bar{S})}(g)) \right|$$

$$= \left| (\hat{\mathbf{ER}}_{\sigma(S)}(f) - \hat{\mathbf{ER}}_{\sigma(S)}(g)) \right| + \left| \hat{\mathbf{ER}}_{\sigma(\bar{S})}(f) - \hat{\mathbf{ER}}_{\sigma(\bar{S})}(g) \right| + \left| \hat{\mathbf{ER}}_{\sigma(S)}(g) - \hat{\mathbf{ER}}_{\sigma(\bar{S})}(g) \right|$$

$$= \left| \hat{\mathbf{ER}}_{\sigma(S)}(g) - \hat{\mathbf{ER}}_{\sigma(\bar{S})}(g) \right| + \left| \frac{1}{m} \sum_{i=1}^{m} (\ell_f(x_i, y_i) - \ell_g(x_i, y_i)) \right| + \left| \frac{1}{m} \sum_{i=m+1}^{2m} \left( \hat{\ell}_f(x_i, y_i) - \hat{\ell}_g(x_i, y_i) \right) \right|$$

$$\leq \left| \hat{\mathbf{ER}}_{\sigma(S)}(g) - \hat{\mathbf{ER}}_{\sigma(\bar{S})}(g) \right| + \frac{1}{m} \sum_{i=1}^{2m} \left| \hat{\ell}_f(x_{\sigma(i), y_{\sigma(i)}}) - \hat{\ell}_g(x_{\sigma(i), y_{\sigma(i)}}) \right|$$

$$< \left| \hat{\mathbf{ER}}_{\sigma(S)}(g) - \hat{\mathbf{ER}}_{\sigma(\bar{S})}(g) \right| + \frac{\eta}{4}.$$

Therefore, we get:

$$\mathbb{P}_{S \sim \mathbb{D}^m} \left[ \left| \hat{\mathbf{ER}}_{\sigma(S)}(f) - \hat{\mathbf{ER}}_{\sigma(\bar{S})}(f) \right| \geq \eta/2 \right]$$

$$\leq \mathbb{P}_{S \sim \mathbb{D}^m} \left[ \left| \hat{\mathbf{ER}}_{\sigma(S)}(g) - \hat{\mathbf{ER}}_{\sigma(\bar{S})}(g) \right| \geq \eta/4 \right]$$

$$\leq |\mathcal{G}| \cdot \max_{g \in \mathcal{G}} \mathbb{P}_{S \sim \mathbb{D}^m} \left[ \left| \hat{\mathbf{ER}}_{\sigma(S)}(g) - \hat{\mathbf{ER}}_{\sigma(\bar{S})}(g) \right| \geq \eta/4 \right].$$

$$\leq \mathcal{N}\left(\frac{\eta}{8}, \hat{\ell}_f, 2m\right) \cdot \max_{g \in \mathcal{G}} \mathbb{P}_{S \sim \mathbb{D}^m} \left[ \left| \hat{\mathbf{ER}}_{\sigma(S)}(g) - \hat{\mathbf{ER}}_{\sigma(\bar{S})}(g) \right| \geq \eta/4 \right]. \qquad \square$$

Our final step is to bound $\mathbb{P}_{\sigma \sim U^m} \left[ \left| \hat{\mathbf{ER}}_{\sigma(S)}(g) - \hat{\mathbf{ER}}_{\sigma(\bar{S})}(g) \right| \geq \frac{\eta}{4} \right]$ for all $(S, \bar{S}) \in (\mathbb{X} \times \mathbb{Y})^{2m}$, which is effected by the last claim.

**Lemma 20.** *For any $f \in \mathcal{F}$, with $\eta \leq 12$:*

$$\mathbb{P}_{\sigma \sim U^m} \left[ \left| \hat{\mathbf{ER}}_{\sigma(S)}(f) - \hat{\mathbf{ER}}_{\sigma(\bar{S})}(f) \right| \geq \frac{\eta}{4} \right] \leq 2 \cdot \exp\left( -\frac{m \cdot \eta^2 \cdot \epsilon}{64H} \right).$$

*Proof.* We use Bernstein's inequality [18] that says for $n$ independent zero-mean random variables $X_i$'s satisfying $|X_i| \leq M$, we have:

$$\mathbb{P}\left( \left| \sum_i^n X_i \right| > t \right) \leq 2 \cdot \exp\left( \frac{-\frac{t^2}{2}}{\sum_{i=1}^{n} \mathbb{E}[X_i^2] + \frac{1}{3}M \cdot t} \right).$$

Note that the quantity $\hat{\mathbf{ER}}_{\sigma(S)}(f) - \hat{\mathbf{ER}}_{\sigma(\bar{S})}(f)$ is simply an average of $m$ random variables, each of which has mean 0 and variance upper bounded by $\frac{H}{\mathbf{ER}_{\mathbb{D}}(f) + \epsilon} \leq \frac{H}{\epsilon}$. Then applying the above bound:

$$\mathbb{P}_{\sigma \sim U^m} \left[ \left| \hat{\mathbf{ER}}_{\sigma(S)}(f) - \hat{\mathbf{ER}}_{\sigma(\bar{S})}(f) \right| \geq \frac{\eta}{4} \right]$$

$$\leq 2 \cdot \exp\left( -\frac{m \cdot \eta^2 \cdot \epsilon}{32H(1 + \frac{\eta}{12})} \right)$$

$$\leq 2 \cdot \exp\left( -\frac{m \cdot \eta^2 \cdot \epsilon}{64H} \right). \qquad \square$$

We will use the Lipschitz property of the loss function to relate the covering numbers of $\ell_{\mathcal{F}}$ and $\mathcal{F}$ as follows:

**Lemma 21.** *Let $\ell : \mathbb{Y} \times \mathbb{Y} \mapsto [0, H]$ be a loss function such that it satisfies:*

$$|\ell(y_1, y) - \ell(y_2, y)| \leq L \cdot |y_1 - y_2|.$$

*Then, for any real valued function family $\mathcal{F}$, we have:*

$$\mathcal{N}\left(\epsilon, \mathcal{F}, m\right) \leq \mathcal{N}\left(\epsilon/L, \ell_{\mathcal{F}}, m\right).$$

*Proof.* Let $S = \{(x_1, y_1) \ldots (x_m, y_m)\} \in (\mathbb{X} \times \mathbb{Y})^m$, and let $g, h \in \mathcal{F}$ be two functions. We have:

$$\frac{1}{m}\sum_{i=1}^{m}|\ell_g(x_i, y_i) - \ell_h(x_i, y_i)| = \frac{1}{m}\sum_{i=1}^{m}|\ell(y_i, g(x_i)) - \ell(y_i, h(x_i))| \leq \frac{L}{m}\sum_{i=1}^{m}|g(x_i) - h(x_i)|.$$

Hence, any $\frac{\epsilon}{L}$ cover for $\mathcal{F}_{|x_1^m}$ is an $\epsilon$ cover for $(\ell_{\mathcal{F}})_{|S}$. $\qquad\square$

Now we are ready for the proof of Lemma 16.

*Proof of Lemma 16.* We have:

$$\mathbb{P}_{S \sim \mathbb{D}^m}\left[\sup_{f \in \mathcal{F}}\left|\mathbf{\hat{ER}}_S(f) - \mathbf{\hat{ER}}_{\mathbb{D}}(f)\right| \geq \eta\right]$$

$$\leq 2 \cdot \mathbb{P}_{(S, \bar{S}) \sim D^m \times D^m}\left[\sup_{f \in \mathcal{F}}\left(\left|\mathbf{\hat{ER}}_S(f) - \mathbf{\hat{ER}}_{\bar{S}}(f)\right| \geq \eta/2\right) \text{ (by Lemma 17)}\right.$$

$$\leq 2 \cdot \sup_{(S, \bar{S}) \in (\mathbb{X} \times \mathbb{Y})^{2m}}\left(\mathbb{P}_{\sigma \sim U^m}\left[\left|\mathbf{\hat{ER}}_{\sigma(S)}(f) - \mathbf{\hat{ER}}_{\sigma(\bar{S})}(f)\right| \geq \eta/2\right]\right) \text{ (by Lemma 18)}$$

$$\leq 2 \cdot \mathcal{N}\left(\eta/8, \hat{\ell}_f, 2m\right) \cdot \max_{g \in \mathcal{G}}\left(\mathbb{P}_{\sigma \sim U^m}\left[\left|\mathbf{\hat{ER}}_{\sigma(S)}(g) - \mathbf{\hat{ER}}_{\sigma(\bar{S})}(g)\right| \geq \eta/4\right]\right) \text{ (by Lemma 19)}$$

$$\leq 4 \cdot \mathcal{N}\left(\eta/8, \hat{\ell}_f, 2m\right) \cdot exp\left(-\frac{m \cdot \eta^2 \cdot \epsilon}{64H}\right) \text{ (by Lemma 20).}$$

Lastly, the function $\hat{\ell}_f(\cdot, \cdot)$ is $L$-Lipschitz in its first argument with $L = \frac{1}{\epsilon \cdot \left(\epsilon + \mathbb{E}_{(x,y) \sim \mathbb{D}}[\ell_f(x,y)]\right)} \leq \frac{1}{\epsilon^2}$. Hence, from Lemma 21, we get that:

$$\mathcal{N}\left(\frac{\eta}{8}, \hat{\ell}_{\mathcal{F}}, 2m\right) \leq \mathcal{N}\left(\frac{\epsilon^2 \eta}{8}, \hat{\ell}_{\mathcal{F}}, 2m\right).$$

$\qquad\square$

Finally, we come to the proof of Theorem 6.

*Proof of Theorem 6.* By Lemma 5, it suffices to show that there exists a learning algorithm $\mathcal{L} : S^m \mapsto \mathcal{F}$ that outputs a function $\hat{f} : \mathbb{X} \mapsto \mathbb{Y} \in \mathcal{F}$ such that $\mathbf{ER}_{\mathbb{D}, \epsilon}(\hat{f}) \leq 3\epsilon$. Recall that in the training phase of Algorithm 4, we use an $\epsilon-\text{SEM}$ algorithm $\mathcal{O}$ that returns a function $\hat{f}$ satisfying:

$$\mathbf{ER}_{S, \epsilon}(\hat{f}) \leq \inf_{\tilde{f} \in \mathcal{F}} \mathbf{ER}_{S, \epsilon}(\tilde{f}) + \epsilon = \epsilon, \tag{1}$$

where the last equality is because $\inf_{\tilde{f} \in \mathcal{F}} \mathbf{ER}_{S, \epsilon}(\tilde{f}) = 0$ in the standard model. So, we are left to bound $\mathbf{ER}_{\mathbb{D}, \epsilon}(\hat{f})$ in terms of $\mathbf{ER}_{S, \epsilon}(\hat{f})$, in particular, that $\mathbf{ER}_{\mathbb{D}, \epsilon}(\hat{f}) \leq 2 \cdot \mathbf{ER}_{S, \epsilon}(\hat{f}) + \epsilon$, which would prove the theorem.

For this purpose, we employ Lemma 16. In this lemma, let us set $\eta = 1/2$ and denote the event

$$\sup_{f \in \mathcal{F}} \frac{\left|\mathbf{ER}_{S, \epsilon}(f) - \mathbf{ER}_{\mathbb{D}, \epsilon}(f)\right|}{\mathbf{ER}_{\mathbb{D}, \epsilon}(f) + \epsilon} \geq \eta = 1/2$$

as the "good" event; if this does not hold, we call it the "bad" event. In the case of the good event, we have

$$\mathbf{ER}_{\mathbb{D},\epsilon}(f) - \mathbf{ER}_{S,\epsilon}(f) \;\leq\; \frac{\mathbf{ER}_{\mathbb{D},\epsilon}(f) + \epsilon}{2}$$

$$\text{i.e.,} \quad \mathbf{ER}_{\mathbb{D},\epsilon}(f) \;\leq\; 2 \cdot \mathbf{ER}_{S,\epsilon}(f) + \epsilon.$$

Plugging in Eq. (1) gives $\mathbf{ER}_{\mathbb{D},\epsilon}(f) \leq 3\epsilon$, as desired.

This leaves us to bound the probability of the bad event, which by Lemma 16, is at most

$$4 \cdot \mathcal{N}_1\left(\frac{\epsilon^2}{16}, \mathcal{F}, 2m\right) \cdot \exp\left(-\frac{m}{256H}\right).$$

This quantity is at most $\delta$ when $m \geq C \cdot \left(\frac{H \log \frac{1}{\epsilon} \log \frac{1}{\delta}}{\epsilon}\right)$ for a large constant $C$, thereby proving the theorem. $\qquad\square$

## A.2 The Agnostic Model: Proof of Theorem 12

First, we give a proof of Lemma 11.

*Proof of Lemma 11.* Let $\chi(\varepsilon) = \min_{f \in \mathcal{F}} \mathbf{ER}_{\mathbb{D},\varepsilon}(f)$ and $\lambda(\varepsilon) = \min_{f \in \mathcal{F}} \mathbf{ER}_{S,\varepsilon}(f)$, where $S \sim \mathbb{D}^m$. For fixed $\hat{y}, y$, we note that $\ell_\epsilon(\hat{y}, y)$ can only decrease when $\epsilon$ increases. Therefore, both $\chi(\varepsilon)$ and $\lambda(\varepsilon)$ are non-increasing with $\varepsilon$.

When $m \geq \frac{H \log \frac{1}{\epsilon} \cdot \log \frac{1}{\delta}}{\epsilon}$, where $\epsilon \leq \varepsilon$, we have from Lemma 16:

$$\mathbb{P}_{S \in \mathbb{D}^m}\left[\sup_{f \in \mathcal{F}}\left(\left|\hat{\mathbf{ER}}_{S,\varepsilon}(f) - \hat{\mathbf{ER}}_{\mathbb{D},\varepsilon}(f)\right| \geq \eta\right)\right] \leq 4 \cdot \mathcal{N}\left(\frac{\eta\varepsilon^2}{8}, \mathcal{F}, 2m\right) \cdot \exp\left(-\frac{m \cdot \eta^2 \cdot \varepsilon}{64H}\right).$$

From Lemma 15, we have : $\mathcal{N}\left(\frac{\eta\varepsilon^2}{8}, \mathcal{F}, 2m\right) \leq \left(\frac{1}{\varepsilon}\right)^{O(d)}$. Setting $\eta = \frac{1}{4}$ and noting that the size of the sample set exceeds $C \cdot \left(\frac{H \cdot d \cdot \log \frac{1}{\epsilon} \cdot \log \frac{1}{\delta}}{\epsilon}\right)$ for some large $C \geq 0$, and small enough $\epsilon \leq \varepsilon$, we claim that with probability $1 - \delta$, we have for all $f \in \mathcal{F}$:

$$\left|\mathbf{ER}_{S,\varepsilon}(f) - \mathbf{ER}_{\mathbb{D},\varepsilon}(f)\right| \leq \frac{\mathbf{ER}_{\mathbb{D},\varepsilon}(f) + \varepsilon}{4} \tag{2}$$

Due to the breaking condition, we have $\varepsilon \geq \mathbf{ER}_{S,\varepsilon}(\hat{f}) \geq \lambda(\varepsilon)$. Then, by Eq. (2), we have:

$$\chi(\varepsilon) \leq \frac{4}{3} \cdot \lambda(\varepsilon) + \frac{\epsilon}{3} \leq \frac{5\varepsilon}{3}.$$

By monotonicity of $\chi(.)$,

$$\chi\left(\frac{5\varepsilon}{3}\right) \leq \chi(\varepsilon) \leq \frac{5\varepsilon}{3}. \tag{3}$$

Also, we have that $\frac{\varepsilon}{2} < \lambda(\frac{\varepsilon}{2}) + \frac{\varepsilon}{6}$. Then, using Eq. (2) again, we get:

$$\lambda\left(\frac{\varepsilon}{2}\right) \leq \frac{5}{4} \cdot \chi\left(\frac{\varepsilon}{2}\right) + \frac{\varepsilon}{8}$$
$$\frac{\varepsilon}{6} \leq \chi\left(\frac{\varepsilon}{2}\right) \leq \chi\left(\frac{\varepsilon}{6}\right).$$

Combining the above with Eq. (3), we get $\frac{1}{6}\varepsilon \leq \Delta_F \leq \frac{5}{3}\varepsilon$. $\qquad\square$

We are now ready to prove Theorem 12.

*Proof of Theorem 12.* From the breaking condition in Algorithm 5 and Lemma 11, we have that $\arg\min_{f \in \mathcal{F}} \mathbf{ER}_{S,\varepsilon}(f) \leq \varepsilon \leq 6 \cdot \Delta_{\mathcal{F}}$. Using the sample error minimization algorithm returns a function $\hat{f}$ such that

$$\mathbf{ER}_{S,\varepsilon}(\hat{f}) \leq \arg\min_{f \in \mathcal{F}} \mathbf{ER}_{S,\varepsilon}(f) + \varepsilon \leq 2\varepsilon$$

Finally, application of Lemma 16 to bound $\mathbf{ER}_{\mathbb{D},\varepsilon}(\hat{f}) = O(\Delta_{\mathcal{F}})$, followed by Lemma 5 gives the desired result. $\qquad\square$

# B    Proofs for the Lower Bounds (Theorems 8 and 13)

In this section, we prove Theorems 8 and 13 which give lower bounds on the sample complexity of an $\varepsilon$-efficient algorithm in the standard and agnostic settings respectively.

## B.1    The Agnostic Model: Proof of Theorem 13

We begin with the agnostic case. We describe a class of distributions $\mathbb{D}_p$ on pairs $(x, y)$, where $p$ is a parameter in $(0, 1)$. Recall that $y$ represents $\log_2 z$, where $z$ is the actual optimal cost of the offline-instance. The distribution $\mathbb{D}_p$ consists of two pairs: $(1, 1)$ with probability $p$, and $(0, 2)$ with probability $1 - p$. Note that the projection of $\mathbb{D}_p$ on the first coordinate is a Bernoulli random variable with probability of 1 being $p$. For the sake of concreteness, the input sequence $\Sigma = \tau_0, \tau_1, \ldots,$ is such that $\text{OPT}(0) = 2$, $\text{OPT}(1) = 4$. The distribution $\mathbb{D}_p$ ensures that the stopping time parameter $T = 0$ with probability $p$, and $T = 1$ with probability $1 - p$. It follows that any online algorithm has only one decision to make: whether to buy the solution for $\mathcal{I}_0$.

Let $\mathcal{A}_p^\star$ be the algorithm which achieves the minimum competitive ratio when the input distribution is $\mathbb{D}_p$, and let $\rho_p^\star$ be the expected competitive ratio of this algorithm. There are only two strategies for any algorithm: (i) buy optimal solution for $\mathcal{I}_0$, and if needed buy the solution for $\mathcal{I}_1$, or (ii) buy the optimal solution for $\mathcal{I}_1$ at the beginning. The following result determines the value of $\rho_p^\star$.

**Lemma 22.** *If $p = \frac{1}{3} + \varepsilon$ for some $\varepsilon \geq 0$, then $\rho_p^\star = \frac{4}{3} - \frac{\varepsilon}{2}$, and strategy (i) is optimal here. In case $p = \frac{1}{3} - \varepsilon$ for some $\varepsilon \geq 0$, then $\rho_p^\star = \frac{4}{3} - \varepsilon$, and strategy (ii) is optimal.*

*Proof.* For strategy (i), the cost of the algorithm is 2 with probability $p$ and 6 with probability $1 - p$. Therefore its expected competitive ratio is

$$p \cdot 1 + \frac{6}{4} \cdot (1 - p) = \frac{3}{2} - \frac{p}{2}.$$

For strategy (ii), the cost of the algorithm is always 4. Therefore, its expected competitive ratio is

$$p \cdot 2 + 1 \cdot (1 - p) = p + 1.$$

It follows that strategy (i) is optimal when $p \geq 1/3$, whereas strategy (ii) is optimal when $p \leq 1/3$. $\square$

We are now ready to prove Theorem 13. Let $\mathcal{A}$ be an algorithm for LTS which is $\varepsilon/4$-efficient with probability at least $1 - \delta$. Further, let $k$ be an upper bound on the sample complexity of $\mathcal{A}$. Given $k$ samples from a distribution $\mathbb{D}_p$, the algorithm outputs a strategy which is a probability distribution on strategies (i) and (ii). We use this algorithm $\mathcal{A}$ to solve the following prediction problem $\mathcal{P}$: $X$ is a random variable uniformly distributed over $\{\frac{1}{3} - \varepsilon, \frac{1}{3} + \varepsilon\}$. Given i.i.d. samples from from 0-1 Bernoulli random variable $T$ with probability of 1 being $X$, we would like to predict the value of $X$.

**Lemma 23.** *Let $\mathcal{A}$ be an algorithm for LTS which is $\varepsilon/4$-efficient with probability at least $1 - \delta$. Then, there is an algorithm that predicts the value of $X$ with probability at least $1 - \delta$ using $k$ i.i.d. samples from $T$.*

*Proof.* Let $t_1, \ldots, t_k$ be i.i.d. samples from $T$. We give $k$ samples $(x_1, y_1), \ldots, (x_k, y_k)$ to $\mathcal{A}$ as follows: for each $i = 1, \ldots, k$, if $t_i = 0$, we set $(x_i, y_i)$ to $(0, 2)$; else we set it to $(1, 1)$. Observe that the samples given to $\mathcal{A}$ are $k$ i.i.d. from the distribution $\mathbb{D}_X$.

Based on these samples, $\mathcal{A}$ puts probability $q_1$ on strategy (i) (and $1 - q_1$ on strategy (ii)). If $q_1 > 1/2$, we predict $X = \frac{1}{3} + \varepsilon$, else we predict $X = \frac{1}{3} - \varepsilon$.

We claim that this prediction strategy predicts $X$ correctly with probability at least $1 - \delta$. To see this, assume that $\mathcal{A}$ is $\varepsilon/4$-efficient (which happens with probability at least $1 - \delta$).

First consider the case when $X = \frac{1}{3} + \varepsilon$. In this case, Lemma 22 shows that the expected competitive ratio of $\mathcal{A}$ is at most $\frac{4}{3} - \frac{\varepsilon}{4}$. As in the proof of Lemma 22, the expected competitive ratio of $\mathcal{A}$ is

$$q_1 \left( \frac{3}{2} - \frac{X}{2} \right) + (1 - q_1)(X + 1).$$

We argue that $q_1 \geq 1/2$. Suppose not. Since $X > 1/3$, $\frac{3}{2} - \frac{X}{2} \leq X + 1$. Therefore, the above is at least (using $q_1 \leq 1/2$ and $X = 1/3 + \varepsilon$)

$$\frac{1}{2}\left(\frac{3}{2} - \frac{X}{2}\right) + \frac{1}{2}(X + 1) > 4/3,$$

which is a contradiction. Therefore $q_1 > 1/2$.

Now consider the case when $X = 1/3 - \varepsilon$. Again Lemma 22 shows that the expected competitive ratio of $\mathcal{A}$ is at most $\frac{4}{3} - \frac{3\varepsilon}{4}$. It is easy to check that if $q_1 \geq 1/2$, then the expected competitive ratio of $\mathcal{A}$ is at least

$$\frac{4}{3} - \frac{\varepsilon}{4},$$

which is a contradiction. Therefore, $q_1 < 1/2$. This proves the desired result. $\qquad\square$

It is well known that in order to predict $X$ with probability at least $1 - \delta$, we need $\Omega\left(\frac{1}{\varepsilon^2}\ln\left(\frac{1}{\delta}\right)\right)$ samples. This proves Theorem 13.

## B.2 The Standard Model: Proof of Theorem 8

Now we consider the standard setting and prove Theorem 8. As in the previous case, the input sequence $\Sigma = \tau_0, \tau_1, \ldots$ will be such that $\mathrm{OPT}(0) = 2$, $\mathrm{OPT}(1) = 4$. Note that the log-cost at the two time-steps are 1 and 2 respectively. Let $\mathbb{X}$ be set of $d$ distinct points (on the real line). Let $\mathcal{F}$ be the set of all $2^d$ functions from $\mathbb{X}$ to $\{0, 1\}$. Clearly, the VC-dimension of $\mathcal{F}$ is given by $d$. For every $f \in \mathcal{F}$, we define a distribution $\mathbb{D}_f$ over pairs $(x, y) \in \mathbb{X} \times \{1, 2\}$ as follows: $\mathbb{D}_f$ is the uniform distribution over $A_f := \{(x, f(x) + 1) : x \in \mathbb{X}\}$. Note that this is an instance of the standard setting, because for any distribution $\mathbb{D}_f$, the corresponding function $f$ maps $x$ to $y$.

Let $\mathcal{A}$ be an algorithm for the LTS problem as above which has expected competitive ratio at most $1 + \varepsilon/4$ with probability at least $1 - \delta$. Let $k$ be an upper bound on the sample complexity of $\mathcal{A}$. The algorithm $\mathcal{A}$, after seeing $k$ samples, outputs a strategy. The strategy gives for each $x \in \mathbb{X}$, a probability distribution over strategies (i) and (ii) as in the previous case.

Now consider the following prediction problem $\mathcal{P}$: we choose a function $f$ uniformly at random from $\mathcal{F}$, and are given $k$ i.i.d. samples from $\mathbb{D}_f$. We would like to predict a function $f' \in \mathcal{F}$ which agrees with $f$ on at least $1 - \varepsilon$ fraction of the points in $\mathbb{X}$.

**Lemma 24.** *Suppose the algorithm $\mathcal{A}$ has the above-mentioned properties. Then given $k$ i.i.d. samples from an instance of $\mathcal{P}$, we can output the desired function $f'$ with probability at least $1 - \delta$.*

*Proof.* Suppose the function $f$ gets chosen. We feed the $k$ i.i.d. samples from $\mathbb{D}_f$ to $\mathcal{A}$. The algorithm $\mathcal{A}$ outputs a strategy $S$ which, for each $x$, gives a distribution $(q_x, 1 - q_x)$ over strategies (i) and (ii).

Given this strategy $S$, we output the desired function $f'$ as follows. For every $x \in \mathbb{X}$, if $q_1(x) \geq 1/2$, we set $f'(x) = 1$, else we set it to 0. We claim that if $\mathcal{A}$ has expected competitive ratio at most $1 + \varepsilon$, then $f'$ agrees with $f$ on at least $\varepsilon$ fraction of points in $\mathbb{X}$.

Suppose not. Suppose $f(x) \neq f'(x)$ for some $x \in \mathbb{X}$. If $f(x) = 0$, then the cost of the optimal strategy here is 2, whereas the algorithm $\mathcal{A}$ follows strategy (ii) with probability at least $1/2$, and its expected cost is more than $2 \cdot \frac{1}{2} + 4 \cdot \frac{1}{2} = 3$. Similarly, if $f(x) = 1$, optimal strategy pays 4. But algorithm $\mathcal{A}$ places at least $1/2$ probability on strategy (i). Therefore, its expected cost is more than $\frac{1}{2} \cdot 4 + \frac{1}{2} \cdot 6 = 5$. In either case, it pays at least 1.25 times the optimal cost. Since $f$ and $f'$ disagree on at least $\varepsilon$-fraction of the points, it follows that the expected competitive ratio of $\mathcal{A}$ (when $x$ is chosen uniformly from $\mathbb{X}$) is more than $1 + \varepsilon/4$, a contradiction.

Since $\mathcal{A}$ has competitive ratio at most $1 + \varepsilon/4$ with probability at least $1 - \delta$, the desired result follows. $\qquad\square$

Now, it is well known that if we want to find a function $f' \in \mathcal{F}$ which matches with $f$ on more than $1 - \varepsilon$ fraction of points in $\mathbb{X}$ with probability at least $1 - \delta$, we need to sample at least $\Omega\left(\frac{d}{\varepsilon}\ln\left(\frac{1}{\delta}\right)\right)$ points from $\mathbb{D}_f$ (see Thm 5.3 in [8]). This proves Theorem 8.

# C Other Omitted Proofs

*Proof of Theorem 1.* Recall that $T$ denotes the length of the input sequence. Let $\tau_i \leq T < \tau_{i+1}$. Then, the cost of the optimal solution, $\text{OPT}(T) \geq \text{OPT}(\tau_i)$ by monotonicity, while the cost of the online solution $\text{SOL}$ is given by:

$$\text{OPT}(\tau_1 - 1) + \text{OPT}(\tau_2 - 1) + \ldots + \text{OPT}(\tau_{i+1} - 1)$$
$$\leq 2 \cdot \text{OPT}(\tau_{i+1} - 1) \leq 4 \cdot \text{OPT}(\tau_i). \qquad \square$$

*Proof of Theorem 2.* When the prediction is correct, i.e., $T = \hat{T}$, the algorithm only runs Phases 1 and 2. At the end of Phase 1, by Theorem 1, the cost of $\text{SOL}$ is at most $4 \cdot \text{OPT}(t_1) \leq (4\epsilon/5) \cdot \text{OPT}(\hat{T})$. In Phase 2, the algorithm buys a single solution of cost at most $(1 + \epsilon/5) \cdot \text{OPT}(\hat{T})$. Adding the two, and noting that the optimal cost is $\text{OPT}(\hat{T})$, we get a consistency bound of $1 + \epsilon$.

For robustness, we consider three cases. First, if $t < t_1$, then the competitive ratio is $4$ by Theorem 1. Next, if $t > t_2$, then the total cost of $\text{SOL}$ is at most $(1 + \epsilon)\text{OPT}(T)$ in Phases 1 and 2 (from the consistency analysis above), and at most $4 \cdot \text{OPT}(T)$ in Phase 3 by Theorem 1. Thus, in this case, the competitive ratio is $5 + \epsilon$. Finally, we consider the case $t_1 \leq t \leq t_2$. Here, the algorithm runs Phases 1 and 2, and the cost of $\text{SOL}$ is at most $(1 + \epsilon) \cdot \text{OPT}(\hat{T})$ by the consistency analysis above. By monotonicity, the optimal solution is smallest when $T = t_1$, i.e., $\text{OPT}(T) \geq \frac{\epsilon}{5} \cdot \text{OPT}(\hat{T})$. Thus, the competitive ratio is bounded by $5\left(1 + \frac{1}{\epsilon}\right)$. $\qquad \square$

*Proof of Theorem 3.* If $T \geq \hat{T}$, the algorithm has to buy a solution that is feasible for $\hat{T}$ at some time $\tau \leq \hat{T}$. In particular, we must have $\text{OPT}(\tau) \leq \epsilon \cdot \text{OPT}(\hat{T})$ for deterministic algorithms, else the consistency bound would be $> 1 + \epsilon$ simply based on being feasible for $t = \tau$ which incurs cost $> \epsilon \cdot \text{OPT}(\hat{T})$ and again for $t = \hat{T}$ which incurs an additional cost of $\text{OPT}(\hat{T})$. This implies a robustness bound of $\Omega\left(\frac{1}{\epsilon}\right)$ if the input $T = \tau$. The same argument extends to randomized algorithms: now, since $\mathbb{E}[\text{OPT}(\tau)] \leq \epsilon \cdot \text{OPT}(\hat{T})$, it follows that $\mathbb{E}\left[\frac{\text{OPT}(\hat{T})}{\text{OPT}(\tau)}\right] \geq \frac{\text{OPT}(\hat{T})}{\mathbb{E}[\text{OPT}(\tau)]} = \Omega\left(\frac{1}{\epsilon}\right)$. $\qquad \square$

*Proof of Lemma 4.* When $T \leq t_1$, the competitive ratio of $4$ follows from the doubling strategy of the algorithm. Next, when $t_1 < T \leq t_2$, the algorithm pays at most $4 \cdot \frac{\epsilon}{5} \cdot \text{OPT}(\hat{T})$ until $t = t_1$ and then pays at most $\left(1 + \frac{\epsilon}{5}\right) \cdot \text{OPT}(\hat{T})$ for the solution $\text{OPT}(t_2)$, which adds up to at most $(1 + \epsilon) \cdot \text{OPT}(\hat{T})$. In contrast, the optimal cost is $\text{OPT}(T)$; hence, the competitive ratio is $(1 + \epsilon) \cdot \frac{\text{OPT}(\hat{T})}{\text{OPT}(T)}$. Finally, when $T > t_2$, then let $\tau_j \leq T < \tau_{j+1}$. The algorithm pays at most

$$\left(1 + \epsilon + 2 + \ldots 2^{j+1}\right) \text{OPT}(\hat{T}) \leq 2^{j+2} \cdot \text{OPT}(\hat{T}),$$

while the optimal cost is at least $2^j \cdot \text{OPT}(\hat{T})$. Hence, the competitive ratio is $\leq 4$. $\qquad \square$

*Proof of Lemma 5.* We note that for all values $y$, the competitive ratio is upper bounded by $1 + \epsilon + 3 \cdot \ell_\epsilon(\hat{y}, y)$, where $\ell_\epsilon(\hat{y}, y)$ is the $\epsilon$-parameterized competitive error of $\hat{y}$. So, the expected competitive ratio is $\leq 1 + \epsilon + 3 \cdot \mathbf{ER}_{\mathbb{D}, \epsilon}(\hat{f})$. $\qquad \square$

*Proof of Lemma 9.* Let the predicted log-cost be $\hat{y} = f_{\mathcal{A}}(x)$. Let $\phi(\hat{y})$ be the sum total of the costs of solutions bought by $\mathcal{A}$ till the optimal log-cost reaches $\hat{y}$. Clearly $\phi(\hat{y}) \geq e^{\hat{y}}$. Since the algorithm $\mathcal{A}$ can be possibly randomized, let $e^{\hat{y}} \leq \phi(\hat{y}) \leq e^{\hat{y}+\epsilon}$ with probability $\alpha$ over the distribution chosen by $\mathcal{A}$.

We define the distribution $\mathbb{D}$ as: $\mathbb{X}$ is just the singleton set $\{x_0\}$ and $\mathbb{Y} = \{\hat{y}, \hat{y} \cdot (1 + \epsilon)\}$. The distribution $\mathbb{D}$ assigns probability $1 - \epsilon$ to $(x_0, \hat{y})$ and $\epsilon$ to $(x_0, \hat{y} \cdot (1 + \epsilon))$ (note that the optimal cost is $e^{\hat{y}}$ and $e^{\hat{y} \cdot (1+\epsilon)}$ in these cases respectively). Note that $\mathbb{D}$ and $f_{\mathcal{A}}$ satisfy $\mathbf{ER}_{\mathbb{D}, \epsilon}(\hat{f}_{\mathcal{A}}) = \epsilon$. The expected competitive ratio of $\mathcal{A}$ is a least

$$\text{CR} \geq (2\alpha + 1 - \alpha) \cdot \epsilon + \alpha \cdot (1 - \epsilon) + (1 - \epsilon) \cdot (1 - \alpha) \cdot (1 + \epsilon) = 1 + \epsilon - (1 - \alpha) \cdot \epsilon^2 \geq 1 + \frac{\epsilon}{2}. \quad \square$$

*Proof of Lemma 10.* Let $\chi(\epsilon) = \min_{f \in \mathcal{F}} \mathbf{ER}_{\mathbb{D},\epsilon}(f)$. Note that, $\chi(\epsilon)$ is non-increasing in $\epsilon$, and $\lim_{\epsilon \to 0} \chi(\epsilon) > 0$. Since $\ell_{\epsilon}(\cdot, \cdot) \leq \frac{5}{\epsilon} - 1$, we have $\chi(2) < 2$. Therefore, there must exist $\Delta_{\mathcal{F}} \in (0, 2)$ such that:

$$\Delta_{\mathcal{F}} = \min_{f \in \mathcal{F}} \mathbf{ER}_{\mathbb{D},\Delta_{\mathcal{F}}}(f)$$

The uniqueness follows from the monotonicity of the function $\chi(.)$ $\qquad\square$

*Proof of Theorem 14.* We show this theorem by using the following lemma:

**Lemma 25.** *Let $\mathcal{A}$ denote Algorithm 4. Then*

$$\mathrm{CR}_{\mathcal{A}}(x, y) = \begin{cases} 4 \text{ when } y \leq \ln \frac{5}{\epsilon} - \hat{y} \text{ or } y > \hat{y} + \ln\left(1 + \frac{\epsilon}{5}\right) \\ (1 + \epsilon) \cdot e^{\hat{y} - y} \text{ otherwise} \end{cases}$$

*Proof.* The proof follows from Lemma 4 by noting that $e^y = \mathrm{OPT}(T), e^{\hat{y}} = \mathrm{OPT}(\hat{T})$, and $\mathrm{OPT}(t_1) = \frac{\epsilon}{5} \cdot e^{\hat{y}}, \mathrm{OPT}(t_2) = (1 + \frac{\epsilon}{5}) \cdot e^{\hat{y}}$. $\qquad\square$

Theorem 14 now follows by noting that the worst case is when $y$ just exceeds $t_1$, i.e., when $y = \ln \frac{5}{\epsilon} - \hat{y}$. $\qquad\square$

# D  Inadequacy of Traditional Loss Functions

In this section, we motivate the use of asymmetric loss function (Definition 6) by showing that an algorithm which uses predictions from a learner minimizing a symmetric loss function, such as absolute loss or squared loss, would have a large competitive ratio. The intuition is that if we err on either side of the true value of $T$ by the same amount, the competitive ratio in the two cases does not scale in the same manner. To formalize this intuition, we define a class of distributions $\mathbb{D}_{\Delta}$, parameterized by $\Delta > 0$, which are symmetric around a real $c$; more concretely this distribution places equal weight on $\{c - \Delta, c + \Delta\}$. Any algorithm relying on a symmetric loss function would always predict $c$. In such a case, the online algorithm $\mathcal{A}$ has no new information. However, if $\Delta$ is large, an offline algorithm is better off buying the solution till $c - \Delta$ first, whereas if $\Delta$ is small, it should buy the solution for $c + \Delta$ in the first step. An algorithm which relies only on $c$ would err in one of these two cases. This idea is formalized in Lemma 26. Our second result (Lemma 27) shows that predicting log-loss within an additive $\epsilon$ factor may result in a $1 + \Omega(\sqrt{\epsilon})$ expected competitive ratio. This further bolsters the case for the loss function as in Definition 6.

**Lemma 26.** *Let $\mathcal{A}$ be an algorithm that uses predictions made by a learner that minimizes symmetric error. Then, one of the following statements is true:*

1. *$\mathbb{E}_{(x,y) \sim \mathbb{D}_{\Delta}}[\mathrm{CR}_{\mathcal{A}}(x, y)]$ is $\Omega(e^{\Delta})$ when $\Delta \geq 4$.*

2. *$\mathbb{E}_{(x,y) \sim \mathbb{D}_{\Delta}}[\mathrm{CR}_{\mathcal{A}}(x, y)] \geq 1 + \Omega(1)$, when $\Delta = \epsilon$, with $\epsilon$ being an arbitrarily small positive real number.*

*Proof.* We define a family of distributions $\mathbb{D}_{\Delta}$, parameterized by $\Delta, 0 \leq \Delta \leq c$, where $c$ is a large enough constant, as follows:

**Definition 12.** *Let $\mathbb{X}$ denote the singleton set $\{x_0\}$ and $\mathbb{Y}$ denote $\{c - \Delta, c + \Delta\}$. The distribution $\mathbb{D}_{\Delta}$ on $\mathbb{X} \times \mathbb{Y}$ assigns probability $\frac{1}{2}$ to both $(x_0, c - \Delta)$ and $(x_0, c + \Delta)$.*

Ideally, we would want an algorithm $\mathcal{A}$ to satisfy $\mathbb{E}_{(x,y) \sim \mathbb{D}_{\Delta}}[\mathrm{CR}_{\mathcal{A}}(x, y)] \to 0$ when $\Delta \to 0$, while still maintaining a worst-case result like $\mathbb{E}_{(x,y) \sim \mathbb{D}_{\Delta}}[\mathrm{CR}_{\mathcal{A}}(x, y)] \leq O(1)$. The following construction shows that using a symmetric loss function would not be helpful. Suppose we use a learner that outputs the function which minimizes a symmetric loss function. Then given samples from $\mathbb{D}_{\Delta}$, such a learner will always yield $\hat{y} = c$ as the prediction.

Since the feature is fixed, the behavior of the algorithm is independent of the feature and hence, it only needs to decide on a list of solutions that it will progressively buy. Let $\tau$ be the cost of the first solution bought by $\mathcal{A}$ that lies inside the interval $[e^{c-\Delta}, e^{c+\Delta}]$ where $\hat{y} = c$ is the predicted log-cost that has been supplied to the algorithm.

There are two possible cases for $\tau$:

(i) $\tau \geq e^c$: With probability $\frac{1}{2}$, the competitive ratio is $\frac{e^c}{e^{c-\Delta}} = e^\Delta$. Hence, $\mathbb{E}\left[\text{CR}_{\mathcal{A}}(x,y)\right] \geq \frac{1+e^\Delta}{2}$. Observe that if $\Delta \geq 4$, then $\mathbb{E}\left[\text{CR}_{\mathcal{A}}(x,y)\right]$ is $\Omega(e^\Delta)$, and hence unbounded.

(ii) $\tau < e^c$: With probability $\frac{1}{2}$, $y = c + \Delta$, in which case the competitive ratio is $\frac{e^c + e^{c+\Delta}}{e^{c+\Delta}} = 1 + e^{-\Delta}$. Therefore, $\mathbb{E}\left[\text{CR}_{\mathcal{A}}(x,y)\right]$ is $1 + \frac{e^{-\Delta}}{2} = 1 + \Omega(1)$, even when $\Delta$ is an arbitrarily small positive $\epsilon$. $\square$

It is worth noting that if we use the loss function as in Definition 6, then Algorithm 4 has expected competitive ratio $1 + O(\epsilon)$ when $\Delta \leq \epsilon$. Further, this algorithm defaults to DOUBLE when $\Delta = \Omega(1)$, and hence has bounded competitive ratio in this case.

We also show that any algorithm which relies on a predictor of log-cost which has an $\epsilon$ bound on the absolute loss must incur $1 + \Omega(\sqrt{\epsilon})$ expected competitive ratio. Comparing this result with Theorem 12 shows that our loss function defined as in Definition 6 gives better competitive ratio guarantees.

**Lemma 27.** *Let $\mathcal{A}$ be a learning-augmented algorithm for* ONLINESEARCH, *that has access to a predictor $P : \mathbb{X} \mapsto [0, H]$ that predicts the log-cost $y$. Moreover, the only guarantee on $P$ is that $\mathbb{E}_{(x,y) \sim \mathbb{D}}\left[|P(x) - y|\right] \leq \epsilon$. Then there is a distribution $\mathbb{D}$ and a predictor $P$ such that* $\mathbb{E}_{(x,y) \sim \mathbb{D}}\left[\text{CR}_{\mathcal{A}}(x,y)\right] \geq 1 + \frac{\sqrt{\epsilon}}{2}$.

*Proof.* Fix an algorithm $\mathcal{A}$. Given the prediction $\hat{y} = 1$, the algorithm outputs a (randomized) strategy for buying optimal solutions at several time steps. Let $\phi$ be the sum total of the costs of the solutions bought by the algorithm before the cost of the optimal solution reaches $e$. Clearly $\phi \geq e$. Let $\alpha$ denote the probability that $\phi \in [e, e^{1+\sqrt{\epsilon}}]$, where the probability is over the distribution chosen by $\mathcal{A}$.

We define the distribution $\mathbb{D}$ as follows: $\mathbb{X}$ is just the singleton set $\{x_0\}$ and $\mathbb{Y} = \{1, 1 + \sqrt{\epsilon}\}$. The distribution $\mathbb{D}$ assigns probability $1 - \sqrt{\epsilon}$ to $(x_0, 1)$ and $\sqrt{\epsilon}$ to $(x_0, 1 + \sqrt{\epsilon})$ (note that the optimal cost is $e$ and $e^{1+\sqrt{\epsilon}}$ in these cases respectively). Note that $\mathbb{E}_{\mathbb{D}}[y]$ is $1 + \epsilon$. Consider the predictor $P$ which outputs the prediction $\hat{y} = 1$, and therefore satisfies the condition $\mathbb{E}_{(x,y) \sim \mathbb{D}}\left[|P(x) - y|\right] \leq \epsilon$. The expected competitive ratio of $\mathcal{A}$ is a least

$$(1 - \sqrt{\epsilon}) \left[\alpha \cdot 1 + (1 - \alpha) \cdot e^{\sqrt{\epsilon}}\right] +$$
$$\sqrt{\epsilon} \left[\alpha \frac{e + e^{1+\sqrt{\epsilon}}}{e^{1+\sqrt{\epsilon}}} + (1 - \alpha) \cdot 1\right]$$

Approximating $e^{\sqrt{\epsilon}}$ by $1 + \sqrt{\epsilon}$, the above expression simplifies to

$$1 + \sqrt{\epsilon} - \epsilon \geq 1 + \frac{\sqrt{\epsilon}}{2}.$$

$\square$