# OpenReview forum: "A Regression Approach to Learning-Augmented Online Algorithms"
_NeurIPS.cc/2021/Conference — NeurIPS 2021 Poster_

### Official Review · Reviewer_2bKw · 2021-07-16

**Rating:** 6
**Confidence:** 1

**Summary:**

This paper focuses on using regression-based ML-predictions in online search problems. In particular, they construct a new loss function that incorporates the asymmetric behavior of the competitive ratio to generate these ML predictions. They show that their proposed algorithm has nearly tight sample complexity bounds under the standard model (when a perfect predition is available in $\mathcal{F}$), and can also achieve good sample complexity guarantees in the agnostic setting.

**Limitations And Societal Impact:**

The authors briefly discussed about limitations and pointed out some future work directions in their conclusion section. I don't have major concerns on this paper, and some minor suggestions are listed in the main review.

**Main Review:**

*Originality / Significance*

This paper is completely out of my field so unfortunately I can't talk much on its originality or signification. My understanding is that the authors propose a new regression approach based on a new loss function to generating ML predictions for online search problems. The overall problem appears quite interesting to me.

*Quality*

The paper tells a complete story. All theoretical results are intuitively explained and they make sense to me. I only have some minor comments listed below.
1. I don't have a good understanding of how restricted these composability and monotonicity assumptions are. Could you intuitively explain how they are used in the analysis and what will fail without them?
2. The loss function looks difficult-to-optimize computationally. How can we get an $\epsilon$-SEM rountine?
3. There seems to be a mismatch between the upper (thm 12) and lower bound (thm 13) in the agnostic setting.

*Clarity*

The paper is clearly written and well organized. I particularly like how the authors motivate their final algorithm step by step.




**Time Spent Reviewing:**

3.5 hours

---

> ### Author Response · Authors · 2021-08-10
> **Response to Reviewer 2bKw**
>
> Many thanks for the comments. We address them below:
> - The composability and monotonicity assumptions are valid in most (but not all) online problems. For instance, these hold for all covering problems (set cover, facility location, network design, scheduling, etc.) as well as for other problems like bin packing, ski rental, etc. An example of a problem where this does not hold is the general Online Convex Optimization (OCO) problem with negative losses. Here, the cost of the optimum solution may decrease over time.
> In terms of how we use them, we need composability to claim that the solution produced by our algorithm (that buys various solutions satisfying subsets of constraints over time) is feasible for the overall instance. Similarly, we need monotonicity because the cost of the algorithm’s solution is a geometric sequence of costs of the optimal solutions that the algorithm buys over time -- without monotonicity, we are not guaranteed such a geometric sequence.
> - We did not explore the computational aspect of optimizing the loss function and instead focused on the sample complexity and generalization bounds. Although there is no immediate general-purpose method for finding an $\epsilon$-SEM for this loss metric, we can use a surrogate loss function depending on the context to arrive at a good approximation and then use non-convex optimization routines.
> - The upper and lower bounds are claiming slightly different things. As for the upper bound: Recall that $\Delta_{F}$ captures the “goodness” of the hypothesis class in predicting the unknown variable (OPT) from the features (x). A small $\Delta_{F}$ implies the existence of a function in class $F$ that captures the relationship between the OPT and the features well. We state our upper bound in terms of $\Delta_{F}$ and give an algorithm with a competitive ratio of $1+O(\Delta_{F})$  (Theorem 12). The lower bound does not use $\Delta_{F}$ for parameterization, but uses $\rho$, which is the competitive ratio of the best algorithm (as defined by definition 4, line 238). In this case, we show that there are cases where one needs $1/\epsilon^2$ samples to have a competitive ratio of $\rho + \epsilon$
> (Theorem 13). Also, the lower bound construction has $d$, $H$ as constants (thus, these do not show up in Theorem 13). It would be interesting to extend the lower bound to general $d$, $H$.

---

### Official Review · Reviewer_Nz3r · 2021-07-16

**Rating:** 7
**Confidence:** 4

**Summary:**

Learning-augmented algorithms have been recently proposed as a way to utilize machine-learned predictions in online algorithm design. This paper considers the related problem of how to learn from past data so that online algorithms can obtain improved performance. Specifically, the paper introduces a general online search problem and design online algorithms for it, both without using predictions and using them. The online search framework is fairly general and encompasses well-studied problems such as generalized ski-rental, load balancing and bin packing. Given a single real-valued predicted parameter, the authors design an algorithm that is $(1+\epsilon)$-consistent and $O(1/\epsilon)$ robust.

Secondly (and more interestingly), the paper designs a learning algorithm to efficiently learn such a parameter given sample access to a data distribution. The authors show that utilizing the structure of the online algorithm, one can define an asymmetric loss function such that the empirical minimizer of this loss function serves as a good prediction for the online algorithm.


**Main Review:**

The paper proposes a natural online problem named as the ''online search’’ problem as a common formalization of online problems where knowledge of the value of the optimal solution suffices to obtain better than worst-case guarantees. Unsurprisingly, a natural algorithm based on the common ``doubling trick’’ gives a competitive ratio of 4. Given access to a single predicted parameter $(\hat{T})$, the proposed Predict-and-Double algorithm gives an essentially optimal tradeoff of $(1+\epsilon)$-consistency and $O(1/\epsilon)$ robustness. Both the design and analysis of this algorithm is fairly natural.

The paper then tackles the important problem of efficiently learning the predicted parameter given only sample access to the data distribution. The paper demonstrates that using traditional loss functions (which are symmetric around the true value) are not sufficient to obtain optimal sample complexity for this setting. Instead, the authors design a specific loss function that is tailored to the behavior of the Predict-and-Double algorithm.

Overall the paper is well-written and easy to read. The paper tackles a timely problem; and the results are non-trivial and interesting.

Minor comments:
- Line 104: The variable k has already been used for the number of options in the previous line.
- Line 108, 118, 138: The Predict-and-Double algorithm referred here has not been defined yet.
- Line 182: A natural idea, then, *is* to progressively …
- Line 195: Did you mean “largest”?


**Time Spent Reviewing:**

2

---

> ### Author Response · Authors · 2021-08-10
> **Response to Reviewer Nz3r**
>
> Many thanks for the comments. We address them below:
> - Multiple uses of $k$: we will change the variable names to prevent any confusion.
> - We will include an explanation for the forward references to PREDICT-AND-DOUBLE.
> - We will fix the missing “is”
> - We will change it to “largest length..”.

---

### Official Review · Reviewer_nzMd · 2021-07-16

**Rating:** 7
**Confidence:** 4

**Summary:**

This paper studies learning augmented algorithms from the perspective of regression.  In particular, this paper studies both how to construct predictions (via regression) and how to use them online for the online search framework.  The online search framework captures problems such as ski rental, online scheduling on identical machines, and online bin packing.  The first main result of this paper consists of tight bounds (up to constants) for robustness and consistency for the online search problem with predictions of the input length.  Next, the paper studies how to predict the value of the optimal solution using regression under the assumption that each instance of the problem has a feature vector x associated with it.  To do this, a novel loss function is proposed that takes into account the competitive ratio of the algorithm utilizing the predicted values.  Nearly tight sample complexity bounds are given for the regression problem.


**Limitations And Societal Impact:**

There is discussion of the limitations of the work, and I do not see any issues in terms of negative societal impact stemming directly from this work.

**Main Review:**

Overall, this is a technically strong and well-written paper which looks at an important problem in learning-augmented algorithms - namely, how should these predictions be constructed.  This paper nicely complements other recent work on this topic - Anand et al. 2020 and Lavastida et al. 2020.

A minor issue is the assumption that each instance of the online problem is associated with a feature vector x that is revealed to the online algorithm up front (in order to use the learned regression model online).  Some discussion as to how to realize this assumption in practice might strengthen the paper.

**Time Spent Reviewing:**

3.5

---

> ### Author Response · Authors · 2021-08-10
> **Response to Reviewer nzMd**
>
> Many thanks for the comments. Typically, learning-augmented online problems come in two flavors: those with offline predictions (like ski rental, makespan minimization, facility location, network design, etc.) and those with online predictions (like caching). Our work focuses on problems of the first kind. Even if the features change over time, they change slowly enough for some problems that one can create a new instance when the features change. For instance, in the case study of an online scheduling problem given in the general comment addressed to all reviewers, the composition of the set of users changes relatively slowly and therefore a new instance of the problem can be created when the feature vector denoting the active users (or share of different user types) changes substantially. Handling predictions that are truly online (like caching) is an interesting direction of future work.

---

### Official Review · Reviewer_8woq · 2021-07-16

**Rating:** 6
**Confidence:** 3

**Summary:**

The paper considers an "online search problem", which includes ski rental and variants as special cases and is also applicable to a classical scheduling problem and online bin packing, and proposes an approach for solving these problems in a learning-augmented setting. The online search problem is very generic: An input sequence is known *offline*, but the actual online input is a prefix of this sequence of unknown length. The goal is to minimize some cost. Some assumptions need to be satisfied, e.g. if A and B are solutions to two subsequences then the combined solution of A and B is feasible for the concatenation of the two subsequences (and the cost of the combined solution is at most the sum of the individual costs). Essentially, this is an abstract version of a generalization of ski rental. Although scheduling and bin packing don't have the property of the online input being a prefix of a known offline input, they can be reduced to the online search problem at the loss of a 1.5 factor in the competitive ratio (the classical competitive ratios for the scheduling and bin packing problems are roughly 1.9 and 1.6 respectively, so a bit higher than those 1.5).

The paper first gives a 4-competitive algorithm for the online search problem (using a standard doubling technique) and a prediction-augmented algorithm with consistency 1+eps and robustness 5+5/eps for any eps (which is pareto-optimal up to a constant factor in the robustness). Here, the prediction is the optimal cost for the online input (or equivalently, the length of the online input). The competitive ratio as a function of the prediction error has an unusual asymmetric shape, which motivates the paper's main contribution:

The main contribution is a regression approach to generate this prediction: It is assumed that for any instance, some feature vector x can be observed at the start, and if z is the optimal value of the instance, then the pair (x,z) comes from some (unknown) probability distribution. Since z is needed as the prediction for the learning-augmented algorithm, the goal of the regression approach is to learn a function f that maps x to z. However, the loss function used for learning is not simply an l1 error or similar, but rather a carefully crafted loss function that serves as a proxy of the competitive ratio as a function the true optimal value z and the predicted value f(x). Such a non-standard loss function is needed because of the asymmetric nature of the competitive ratio as a function of the prediction error. Upper and lower bounds on the sample complexity are given so that the resulting algorithm has competitive ratio close to 1+eps with high probability, where epsilon (roughly speaking) is the loss of the best function f from the considered function class.

**Ethical Concerns:**

No ethical concerns.

**Limitations And Societal Impact:**

Limitations yes, potential negative societal impact no (checklist says that they are not aware of potential negative societal impact)

**Main Review:**

The main contribution of the paper seems original and the regression approach may be applicable to other problems, especially when the competitive ratio as a function of the prediction error is asymmetric. For this reason, the paper's contribution has the potential of being significant, but it is hard to judge whether it really is. Some case study might be useful: Does the approach actually lead to practical performance gains for any of the considered problems mentioned in the introduction? How would the feature vector x and the hypothesis class of possible functions f be chosen? For scheduling and online bin packing I do not personally expect much/any improvement because the 1.5 factor loss due to the reduction is already close to the classical competitive ratio, and loss due to prediction error will come on top of this.

The presentation is mostly clear and overall this paper is a candidate for acceptance at Neurips.

Minor comments:
- Line 118 and 138 it is hard to understand what you mean by "tries to buy an OPT" as we have not read yet what those algorithms do.
- It was not completely clear to me whether for bin packing you still also need a prediction of the number of items of size between 1/2 and 2/3. If I understand correctly, then the prediction of OPT’ serves as a substitute for this?
- The order of the arguments y and hat{y} in Definition 6 is different from how it is used in Definition 7.


POST-REBUTTAL COMMENTS:
I appreciate the example given for what could be used as a feature vector. It is still unclear whether it would lead to practical improvement though. The authors are right that the state-of-the art (in terms of worst-case guarantee) algorithms for bin packing and scheduling are very complicated, but there also exist very simple algorithms with similar guarantees (e.g. BestFit/FirstFit for Bin Packing are 1.7-competitive for Bin Packing and typically much better in practice), so I remain skeptical about practical benefits for these problems. Overall, I retain my slightly positive score.

**Time Spent Reviewing:**

4

---

> ### Author Response · Authors · 2021-08-10
> **Response to Reviewer 8woq**
>
> Many thanks for the comments.
> We give a case study illustrating the use of our framework in the general comment addressed to all reviewers.
> For problems like bin packing and scheduling,  we improve on worst case competitive ratios using a simple greedy strategy augmented with our PREDICT-AND-DOUBLE meta-algorithm. In contrast, the state of the art online algorithms for these problems are very complicated. In practice, both algorithms are likely to perform better than their worst case bounds, but we believe  the simplicity of our algorithm gives it a significant edge in terms of real world implementability and performance.
> Response to minor comments:
> - Line 118 and 138: We will clarify the forward reference by explicitly stating that the PREDICT-AND-DOUBLE algorithm buys the optimal solution for various prefixes of the input sequence, and we replace these steps with buying an approximately optimal solution instead.
> - Yes, if we have a prediction on OPT’, we do not require a prediction on the number of items of moderate size. We will clarify this in the paper.
> - The order of the arguments $y$ and $\hat{y}$: This is a typo that we will fix.

---

### Author Response · Authors · 2021-08-10
**General comment for all reviewers**

We thank all the reviewers for their helpful comments. In this general comment, we give a case study illustrating the use of our framework. We also give separate replies to each reviewer answering their individual questions and comments.
Consider the problem of balancing the load between machines/clusters in a data center where remote users are submitting jobs. The goal is to minimize the maximum load on any machine, also called the makespan of the assignment. The optimal makespan, which we would like to predict, depends on the workload submitted by individual users who are currently active in the system. Therefore, we would like to use the user features to predict their behavior in terms of the workload submitted to the server. A typical feature vector would then be a binary vector encoding of the set of users currently active in the system, and based on this information, a learning model trained on historical behavior of the users can predict  (say) a histogram of loads that these users are expected to submit, and therefore, the value of the optimal makespan. The feature space can be richer, e.g., including contextual information like the time of the day, day of the week, etc. that are useful to more accurately predict user behavior. Irrespective of the precise learning model, the main idea in this paper is that the learner should try to optimize for competitive loss instead of standard loss functions. This is because the goal of the learner is not to accurately predict the workload of each user, but to eventually obtain the best possible makespan. For instance, a user who submits few jobs that are inconsequential to the eventual makespan need not be accurately predicted. Our technique automatically makes this adjustment in the loss function, thereby obtaining better performance on the competitive ratio.

---

### Decision · Program_Chairs · 2021-09-27

**Decision:**

Accept (Poster)

**Comment:**

This work makes progress in the recently popular "learning-augmented algorithms" framework and argues that a regression-based approach forms a correct learning solution for many of the problems. The authors do a complete study of this question; of particular interest is their exploration of the learning problem itself. Here they show that special care must be taken in setting up the loss function for the regression problem to achieve good results. Overall, a solid contribution.